# Improvements in health-related quality of life are maintained long-term in patients prescribed medicinal cannabis in Australia: The QUEST Initiative 12-month follow-up observational study

**Margaret-Ann Tait**[1]*, **Daniel S.J. Costa**[2], **Rachel Campbell**[2], **Leon N. Warne**[3,4,5], **Richard Norman**[6], **Stephan Schug**[7], **Claudia Rutherford**[1]

1 Faculty of Medicine and Health, The University of Sydney, Sydney Nursing School, Sydney, New South Wales, Australia, 2 Faculty of Science, The University of Sydney, School of Psychology, Sydney, New South Wales, Australia, 3 Little Green Pharma, West Perth, Western Australia, Australia, 4 College of Science, Health, Engineering and Education, Murdoch University, Perth, Western Australia, Australia, 5 School of Pharmacy and Biomedical Sciences, Curtin Health Innovation Research Institute, Curtin University, Perth, Western Australia, Australia, 6 School of Population Health, Curtin University, Perth, Australia, 7 Medical School, University of Western Australia, Perth, Australia

* margaret-ann.tait@sydney.edu.au

## Abstract

### Aims

Since 2016, more than one million new patients with chronic health conditions have been prescribed medicinal cannabis in Australia. We aimed to assess overall health-related quality of life (HRQL), pain, fatigue, sleep, anxiety, depression, and motor function in a large real-world sample of patients prescribed medicinal cannabis. We previously found all patient-reported outcomes improved in the first 3-months and hypothesised that improvements would be maintained to 12-months.

### Methods

The QUEST Initiative, a multicentre prospective study, recruited adult patients with any chronic health condition newly prescribed medicinal cannabis oil between November 2020 and December 2021. Participants identified by 114 clinicians across Australia completed validated questionnaires at baseline, then 2-weeks titration, and 1-,2-,3-,5-,7-,9- and 12-months follow-up.

### Results

Of 2744 consenting participants who completed baseline assessments, 2353 also completed at least one follow-up questionnaire and were included in analyses, with completion rates declining to 778/2353 (38%) at 12-months. Ages ranged between 18–97 years (mean 50.4y; SD = 15.4), 62.8% were female. Chronic conditions commonly treated included musculoskeletal pain (n = 896/2353; 38.1%), neuropathic pain (n = 547/2353;

**Data availability statement:** All data files are available from the Sydney eScholarship database (https://url.au.m.mimecastprotect.com/s/7U2CCzvkyVC84v8AOs4gfPO?domain=hdl.handle.net).

**Funding:** The University of Sydney received funding from Little Green Pharma Ltd. to support CR and MT to conduct this study. The funder played no role in the study design; in the collection, analysis, and interpretation of data; in the writing of the report; nor in the decision to submit the article for publication. The study was independently investigator-led and all authors had full access to all data (including statistical reports and tables) in the study and take responsibility for the integrity of the data and the accuracy of the data analysis.

**Competing interests:** I have read the journal's policy and the authors of this manuscript have the following competing interests: The University of Sydney received funding from Little Green Pharma Pty Ltd. to support MT and CR to conduct the submitted work; LW is a paid employee of Little Green Pharma Pty Ltd.; no other relationships or activities that could appear to have influenced the submitted work. All authors have completed the ICMJE uniform disclosure form. This does not alter our adherence to PLOS ONE policies on sharing data and materials.

**Abbreviations:** CBD, Cannabidiol; DASS21, 21 item Short Form for Depression Anxiety and Stress Scale; EORTC, European organisation for research & treatment of cancer; EQ-5D, EuroQol five-dimension scale for measuring generic health status; HRQL, Health-related quality of life; LGP, Little green pharma Ltd; MC, Medicinal cannabis; PROM, Patient-reported outcome measure; PROMIS, Patient-reported outcomes measurement Information system; PRO, Patient reported outcomes; QLQ-C15, 15 item version of QLQ-C30 for palliative care patients; QLQ-C30, Generic quality of life questionnaire for cancer patients developed by the EORTC; RCT, Randomized controlled clinical trials; SD, Standard deviation; TGA, Australian therapeutic goods administration; THC, delta-9-tetrahydrocannabinol; USD, United States dollar

23.2%), insomnia (n=546/2353; 23.2%), anxiety (n=520/2353; 22.1%), and mixed depressive and anxiety disorder (n=263/2353; 11.2%). Clinically meaningful improvements were observed in HRQL: EQ-5D-5L index (d=0.52) and QLQ-C30 summary scores (d=0.91), PROMIS fatigue (d=0.51) and sleep disturbance (d=0.76). Participants diagnosed with chronic pain experienced clinically meaningful improvement in scores on QLQ-C30 pain (d=0.5), PROMIS pain intensity (d=0.76), and PROMIS pain interference (d=0.76). There was significant improvement in DASS anxiety (d=0.69) and DASS depression (d=0.65) for those with anxiety or depressive conditions, but no motor function improvements observed for participants with movement disorders. All observed improvements were statistically significant.

## Conclusions

Statistically significant and clinically meaningful improvements in overall HRQL, fatigue, and sleep disturbance were maintained over 12-months in patients prescribed medical cannabis for chronic health conditions. Anxiety, depression, insomnia, and pain also improved over time for those with corresponding health conditions.

## Study registration

Australian New Zealand Clinical Trials Registry: ACTRN12621000063819

## Introduction

Almost half of the Australian population suffers from chronic health conditions,[1] with an estimated 3.6 million living with chronic pain,[2] 3.3 million with anxiety disorders, and 1.5 million with sleep disorders,[3] all negatively impacting their Health-Related Quality of Life (HRQL). Research into the therapeutic benefits of medicinal cannabis (MC) has increased since discovery of the analgesic properties in cannabis plant compounds, delta-9-tetrahydrocannabinol and cannabidiol (THC and CBD),[4] and fuelled by growing concerns around opioid misuse and adverse events,[5] including bowel dysfunction, cognitive decline, endocrinopathy, hospitalization, and death from overdose [6]. With support from the community, advocacy groups lobbied the Australian government to bring about legislation changes in 2016, [7] which allows patients not responding to conventional treatment to access MC with a prescription from clinicians regulated by the Therapeutic Goods Administration (TGA). TGA records to date show that more than one million new patients in Australia have received MC prescriptions, [8] for over 200 health conditions [9].

Regarded as the gold standard for assessing pain, [10] a patient-reported outcome (PRO) is any report about health status that comes directly from patients, [11] and is important when evaluating the impact of new treatments for chronic health conditions where the main goal is to alleviate symptoms [12]. HRQL is a PRO that encompasses the overall impact of disease or treatment across areas such as physical, emotional, social, and cognitive function, as well as bodily discomfort and symptoms like pain [13]. Regulatory bodies on safety and quality in health care in Australia [14] and those overseeing medical research funding, health service delivery, and product labelling internationally, [11,15–17] often require evidence gathered using validated PRO measures (PROMs) to assess the value of treatment. To better inform regulation and policymaking, evidence from patients prescribed MC in clinical practice is needed to evaluate change in HRQL and other PROs in the real-world [18, 19].

The QUEST initiative (**QU**ality of life **E**valuation **ST**udy) assessed patient-reported HRQL, pain, fatigue, sleep disturbance, anxiety, depression, and motor function in patients with chronic health conditions prescribed MC in Australia. Our short-term results, reported elsewhere,[20] found that within the first three months of MC therapy, participants reported clinically meaningful improvements in HRQL, fatigue, and sleep disturbance, and in health conditions associated with anxiety, depression, and pain. It is typical when evaluating clinical care outcomes in chronic conditions to assess 12-month follow-up [21]. This study aimed to assess 12-month follow-up data to determine if our previously reported improvements at 3-months were maintained long-term and to explore differences across health conditions and MC compositions. We hypothesised that improvements in PROs from baseline would be maintained long-term in patients prescribed MC, and that patients with specific conditions would have sustained improvements in condition-specific symptoms.

## Methods

The STROBE statement for reporting observational studies was followed [22]. Ethical approval was granted by University of Sydney Human Research Ethics Committee (HREC) Project#:2020/589 and informed written consent to participate in the study was obtained from all participants. This study was registered with the Australian New Zealand Clinical Trials Registry: ACTRN12621000063819. Full details of study design, eligibility, recruitment procedures, and data collection are reported in the published study protocol [23].

### Study population and design

The QUEST initiative is a multicentre prospective study of patients with chronic health conditions newly prescribed MC across Australia between 27 November 2020 and 23 December 2021. All clinicians prescribing Little Green Pharma (LGP) MC oil products across Australia were informed of the study and invited to contact the researchers to receive the study training and information required to screen their patients for eligibility. Clinicians entered clinical information on eligibility screening forms via the web-based research data capture system, REDCap [24]. Eligibility for the 12-month follow-up study included patients ≥18 years old with prescriptions for LGP MC oil products, and able to read and self-complete online PROMs in English. To achieve a pre-therapy baseline, patients were excluded if they had accessed prescribed MC within the previous 4-weeks; selected because it ensured the minimum wash-out period of 13–30 days had passed, [25, 26] and was greater than the maximum recall period of PROMs used in the study. Palliative care patients were identified by clinicians following the ICD-11 definition of having a life expectancy of only a few months [27]. Accordingly, PROMs were only administered to palliative care patients for the first 3-months of the QUEST study, excluding them from the 12-month analysis. Our 3-month findings for participants receiving end of life palliative care are reported elsewhere [20]. Invitations were emailed to eligible patients directly from REDCap. All participants purchased LGP products at the same price of AUD$150 (USD $98) per 50ml bottle, standardised to allow future health economic evaluation. Depending on individual dosing needs, each 50ml bottle typically lasts between 6–12 weeks. The four LGP products contained the following ratios of THC and CBD dissolved in a medium chain triglyceride carrier oil: LGP Classic 1:20 (1mg THC and 20mg CBD per ml), LGP Classic 10:10 (10mg THC and 10mg CBD per ml), LGP Classic 20:5 (20mg THC and 5mg CBD per ml), LGP Classic CBD 50 (50mg CBD per ml).

### Data collection

Clinicians completed basic patient demographics (age and sex), clinical characteristics, and selected up to two health conditions that were being treated with MC. Informed written

consent, further demographics, and PROMs were completed electronically by participants. All PROMs were completed at baseline prior to commencing MC therapy, after approximately 2-weeks titration (optimal benefit of therapy was expected to be achieved 2-weeks after commencing therapy), monthly for 3 months, then at 5-, 7-, 9-, and 12-months post titration. Follow-up assessment timepoints were selected to align with TGA guidance for MC monitoring, [28] and clinical guidelines [29, 30]. At each assessment timepoint, participants received automated reminders to complete PROMs within 7-days, with non-responders recorded as missed assessments. Data collection ended 19 March 2023.

## PROMs

We assessed PROs using validated PROMs for HRQL (EQ-5D-5L Index; QLQ-C30 Summary score), pain (QLQ-C30-pain subscale), fatigue (PROMIS fatigue 13a), sleep (PROMIS sleep disturbance 8b), anxiety (DASS-anxiety scale), and depression (DASS-depression scale). A description and justification for each PROM administered to all participants is reported in detail elsewhere [20,23]. Additional PROMs for pain (PROMIS pain intensity 3a; PROMIS pain interference 8a) and motor function (Neuro-QoL Upper extremity function) were administered to participants with diagnosed chronic pain conditions or movement disorder in this 12-month study and are described in S1 Table.

## Statistical analyses

**Statistical considerations.** Participants' PROs were included in the analyses if they had a score at baseline and at least one follow-up assessment. Our target sample size of 2142 was determined *a priori* with power to detect the smallest QLQ-C30 effect size threshold [31] using a two-sided significance level of 1%, as reported in the study protocol [23]. All PROMs were scored following instructions provided by the PROM developers. The HealthMeasures Scoring Service [32] calculated PROMIS measure T-scores with a mean of 50 and standard deviation of 10 in the general population (US 2000 Census) for assessing pain, fatigue, and motor function, and in combination with a clinical sample for assessing sleep disturbance [33]. EQ-5D responses were transformed using the most recent Australian population utility weights [34] and combined to produce a health index ranging from 0 (death) to 1 (perfect health).

Means, standard deviations (SD), and standardized mean change from baseline (Cohen's d) with 95% confidence intervals were calculated for each assessment timepoint. Linear mixed models were used to examine change over time in PRO scores (including baseline to 3-months), with time included as a random factor.

The model adjusted for PRO response levels at baseline and sex, with duration of pain and age modelled as fixed factor covariates. Participant age, sex, and duration of pain condition, were previously identified as significant covariates in this cohort [20]. Change over time was analysed by looking at linear and quadratic trends to determine whether there was constant change over time (linear) or change at a changing rate (linear + quadratic); and baseline was compared to mean of all follow-up scores when assessing differences between groups. Averaging follow-up was justified by results from the 3-month analyses, which demonstrated that the largest changes occurred shortly after MC-therapy initiation with minimal change thereafter [20]. Paired sample t-tests compared mean change from baseline to each follow-up timepoint. In addition, the average follow-up scores for DASS anxiety and depression subscales were coded into severity categories and compared to the distribution of severity categories at baseline using a One-Sample Chi-squared test. Statistical significance, defined as p-value < 0.05, was Hochberg-adjusted to account for multiple comparisons, [35] and analyses conducted with IBM SPSS Statistics 28.0 program.

Findings from our 3-month study revealed that participants were often prescribed more than one product,[20] therefore total daily dosage of THC and CBD was calculated and grouped into four of the five active ingredient categories used by the TGA:[36] CBD-only (CBD ≥ 98%); CBD-dominant (CBD ≥ 60% and < 98%); CBD:THC-balanced (CBD < 60% and ≥ 40%); and THC-dominant (THC 60–98%); with the fifth TGA category, THC-only (THC ≥ 98%), not available in this study.

**Clinically meaningful change.** The primary focus of our analyses was to examine clinically meaningful change in PROs. This is important because our large sample size provided power to detect small changes as statistically significant that may not actually be regarded as clinically relevant or important to patients. Minimally clinically important differences (MCIDs) [37] in PROs over time were evaluated using existing guidelines where available. The MCID for EQ-5D-5L index score falls between 0.037 and 0.069 in general populations, [38] and evaluating meaningful within-group change using PROMIS measures is between 2 and 6 with consensus on 3 T-score points [39, 40]. The MCID for DASS-21 depression and anxiety scales is change of 5 points from one severity category to another [41]. The recommended QLQ-C30 pain subscale MCID is 5 points, [31] however there are currently no published MCIDs for the QLQ-C30 Summary Score.

In the absence of guidelines, Cohen's d = 0.5 was used for the QLQ-C30 Summary Score MCID. This threshold of half of the standard deviation of change score has previously been determined as suitable for discriminating HRQL change in chronic diseases, [42] and is reported for all PROs.

## Patient and Public Involvement

Patient participants voluntarily provided self-rated PROM responses. Participants received a summary of findings at the end of the study but were not directly involved in developing the research question or study design.

## Results

Of 3302 invited eligible patients by 114 clinicians, 2744 (83%) provided consent and completed baseline PROMs and demographics. Of those, 2353 (86%) completed at least one follow-up PROM and were included in the analysis (Fig 1). During study follow-up, 322 (11.7%) enrolled participants withdrew due to lack of therapeutic benefit (132, 41%), finding an alternative treatment (70, 22%), unwanted side-effects (64, 20%), and financial cost (57, 17%). PROM completion rates for participants remaining on the study at each follow-up ranged from 82.8% at 1-month to 38% at 1-year. The 391 who dropped out after only completing baseline without providing a reason were generally younger, male, less educated, and less likely to be married, than those who continued on the study (Table 1). When looking at the 2353 participants who continued, those still remaining at 12-months were slightly older and less likely to have been diagnosed with an anxiety disorder.

Participants were aged between 18-97 years (mean 50.4y; SD = 15.4), 62.8% female, 37.4% University educated, and more than a quarter were either unemployed, on leave, or on limited work duties, due to their poor health (Table 1). S2 Table provides additional demographic information on gender identity and ethnicity.

The range and proportion of conditions being treated were similar for participants included in the analysis compared with those who dropped out at baseline (S3 Table). Half of participants were prescribed MC for more than one condition (n = 1244/2353; 53%), with the majority treated for chronic pain conditions (n = 1615/2353; 68.6%). Other common conditions included insomnia (n = 546/2353; 23.2%), anxiety (n = 520/2353; 22.1%),

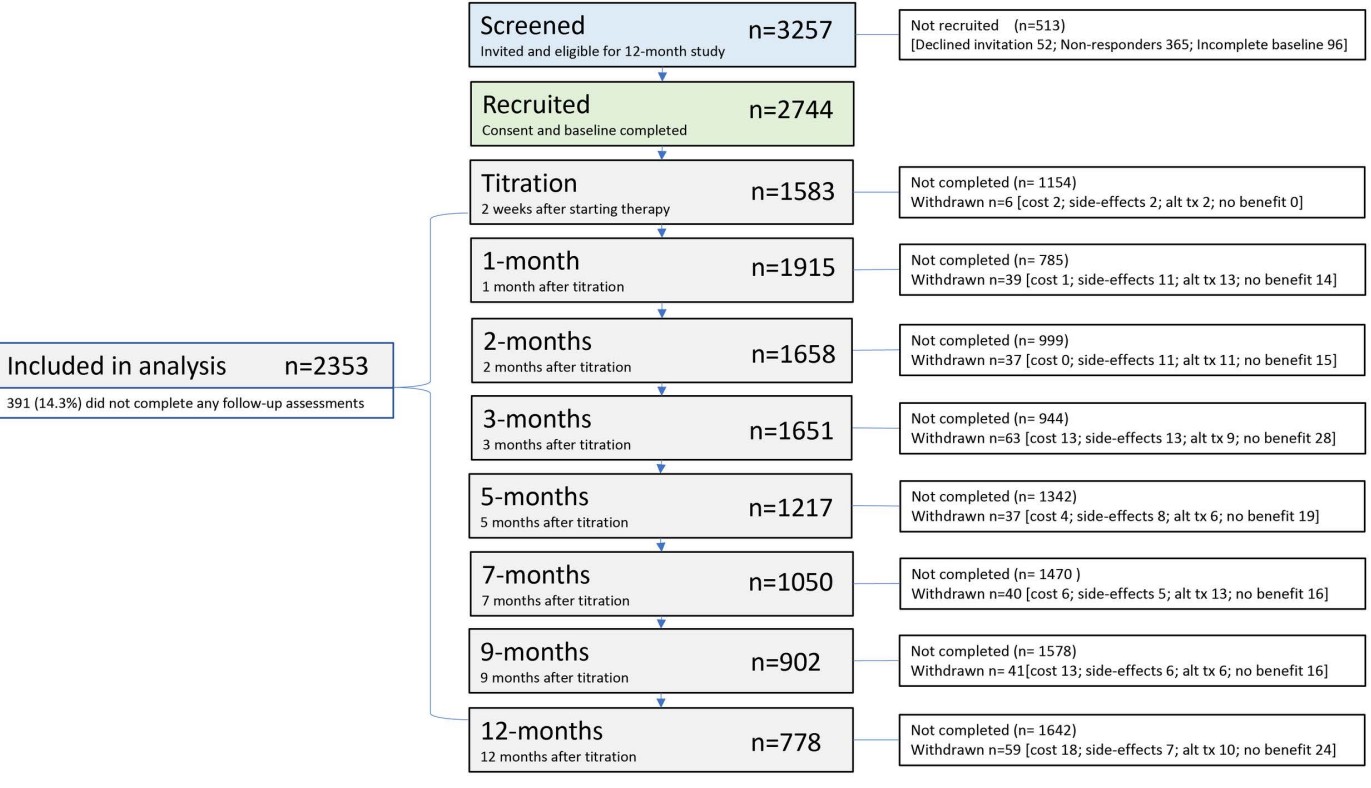

**Fig 1. Study Recruitment Flow.**

and mixed anxiety and depression (n = 263/2353; 11.2%). Ninety participants had a cancer diagnosis (receiving cancer-treatment), of which only 28 were prescribed MC for cancer-related pain.

Within each MC composition category, the median daily doses were: CBD-only – 50mg (IQR: 25, 100) equivalent to 1ml LGP Classic CBD; CBD-dominant – 30mg CBD (IQR: 15, 55) and 3mg THC (IQR: 1, 8) similar to 1.5 ml LGP Classic 1:20; CBD:THC-balanced – 7.5mg CBD (IQR: 3, 15) and 7.5mg THC (IQR: 3, 15) equivalent to 0.75ml LGP Classic 10:10; and THC-dominant – 5mg CBD (IQR: 2, 10) and 20mg THC (IQR: 8, 30) equivalent to 1ml LGP Classic 20:5. The number of participants at each follow-up timepoint taking the different combinations of active ingredients are shown in Fig 2.

Less than 5% (109/2353) of participants had been prescribed MC previously (but not within 4-weeks prior to joining the study), and 576/2353 (24.5%) had used cannabis recreationally, or medicinally without a prescription, within 12-months prior to joining. At baseline, two-thirds of participants (1621/2353) were taking medications other than MC on a regular daily basis to manage their condition, of which 488 (30%) were opioids, with an additional 38 participants taking opioids occasionally. In total, 526 (22.6%) participants were regularly taking opioids daily or as needed when they joined the study. During the study 1100 (47%) participants reported that due to taking MC they had reduced their use of at least one of their other prescribed medications to manage their symptoms. Of these, 526 (48%) had completely stopped taking one or more medications due to taking MC. By the end of the 12-month follow-up period, 370/526 (70%) participants had reduced or stopped their opioid medications.

**Table 1. Baseline characteristics of QUEST participants grouped by participants who completed baseline PROMs only, and those included in the 12-month analyses (completed baseline plus at least one follow-up).**

| Characteristics | Completed baseline only | Included in analysis | P value ($X^2$) |
|---|---|---|---|
| Total (*n = 2744*) | 391 | 2353 | |
| **Age (years),** *mean (SD)* | 47.7 (17.1) | 50.4 (15.4) | **0.002** |
| **Sex,** *n (%)* | | | |
| Male | 157 (40.2) | 874 (37.1) | 0.230 |
| Female | 232 (59.3) | 1477 (62.8) | |
| Indeterminate/Intersex | 1 (0.3) | 2 (0.1) | |
| **Living arrangements,** *n (%)* | | | |
| Live alone | 76 (19.4) | 479 (20.4) | 0.124 |
| Live with partner | 226 (57.8) | 1423 (60.5) | |
| Live with carer | 10 (2.6) | 41 (1.7) | |
| Live with other | 74 (18.9) | 401 (17) | |
| Live in assisted care home | 3 (0.8) | 7 (0.3) | |
| Missing | 2 (0.5) | 2 (0.1) | |
| **Marital Status,** *n (%)* | | | |
| Single | 96 (24.6) | 525 (22.3) | **0.032** |
| Married | 169 (43.2) | 1106 (47) | |
| Separated | 22 (5.6) | 92 (3.9) | |
| Divorced | 35 (9.0) | 252 (10.7) | |
| Widowed | 20 (5) | 72 (3.1) | |
| Cohabitating | 47 (12.0) | 304 (12.9) | |
| Missing | 2 (0.5) | 2 (0.1) | |
| **Work Status,** *n (%)* | | | |
| Full time | 138 (32.2) | 658 (28.6) | 0.792 |
| Part time | 68 (15.9) | 361 (15.7) | |
| At work but limited hours/duties | 27 (6.3) | 131 (5.7) | |
| Retired | 68 (15.9) | 449 (19.5) | |
| Unemployed due to illness | 78 (18.2) | 394 (17.2) | |
| Unemployed NOT due to illness | 6 (1.4) | 43 (1.9) | |
| On leave due to illness | 10 (2.3) | 57 (2.5) | |
| Home duties | 15 (3.5) | 99 (4.3) | |
| Studying only | 9 (2.0) | 65 (2.8) | |
| Voluntary work | 6 (1.4) | 25 (1.1) | |
| Retraining | 4 (0.9) | 15 (0.7) | |
| Missing | 6 (1.4) | 30 (1.3) | |
| **Education,** *n (%)* | | | |
| Primary School | 6 (1.5) | 23 (1.0) | **0.033** |
| High School | 120 (30.7) | 599 (25.5) | |
| Certificate or Diploma | 129 (33.0) | 850 (36.1) | |
| University or higher | 134 (34.3) | 879 (37.4) | |
| Missing | 2 (0.5) | 2 (0.1) | |

SD standard deviation; $X^2$ Pearson's chi-squared; P values in bold are significant.

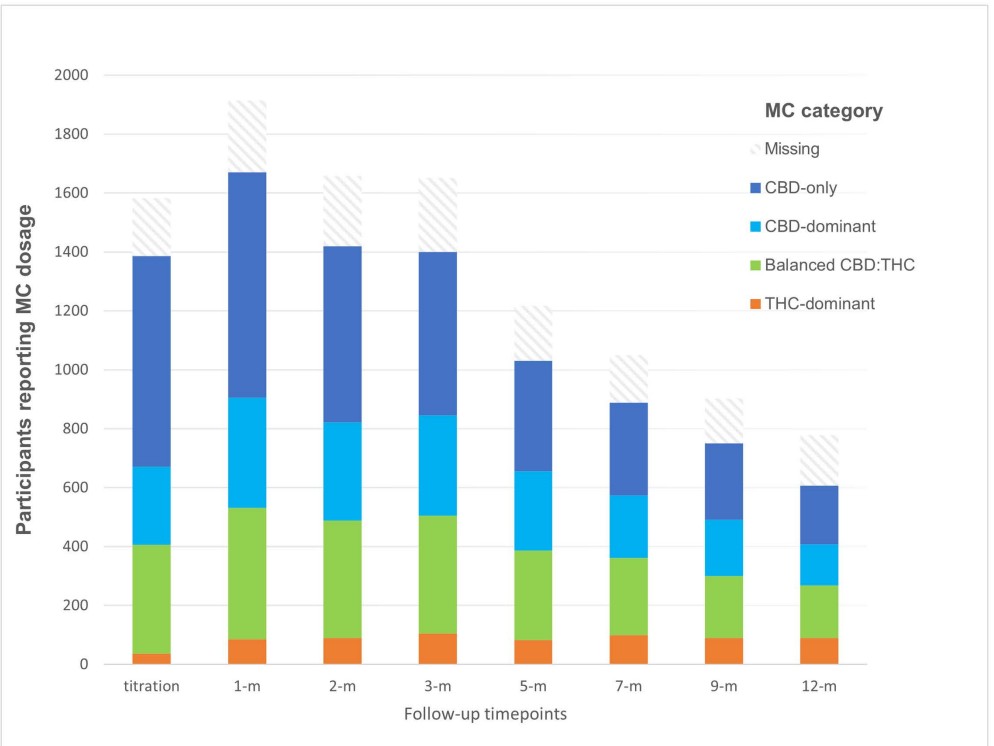

**Fig 2. Number of participants at each follow-up timepoint taking MC doses containing average daily CBD to THC ratios within the four categories of active ingredients.**

## PROs

Results in Table 2 show mean differences in HRQL, pain, sleep, fatigue, depression, and anxiety scores across the whole cohort from baseline to 5-, 7-, 9-, and 12-months follow-up and clinically meaningful significance of effect sizes (results for first three months of the QUEST study are reported elsewhere[20]).

## HRQL

EQ-5D-5L index scores (n = 2353) displayed significant linear ($t_{(9028)}$ = 9.79, p < 0.001) and quadratic ($t_{(13037)}$ = -10.05, p < 0.001) trends over time, signifying a large initial improvement maintained thereafter (Fig 3a). Adjusted scores improved on average by 0.114 (SD = 0.219; 95%CI:0.111, 0.122) from 0.625 (SD = 0.240) at baseline to mean follow-up of 0.739 (SD = 0.224), indicating a clinically meaningful improvement (d = 0.52) greater than the recommended MCID [45].

QLQ-C30 summary scores (n = 2353) also showed significant linear ($t_{(2351)}$ = 17.32, p < 0.001) and quadratic ($t_{(5595)}$ = 19.46, p < 0.001) trends, suggesting initial improvement maintained over 12-months(Fig 3b). The adjusted mean difference from 58.92 (SD = 16.7) at baseline to 69.63 (SD = 17.66) mean follow-up was 10.71 (SD = 11.77) indicating a clinically meaningful improvement (d = 0.91; 95%CI: 0.85, 0.97).

Table 3 reports change in HRQL from baseline to mean post-MC therapy across different health conditions, for participants having only one of the listed conditions.

**Table 2. Change in HRQL, pain, sleep, fatigue, depression, and anxiety from baseline to 5-, 7-, 9-, and 12-months post-titration in 2353 participants with any health condition prescribed medical cannabis.**

| Outcome | PROM | Follow-up timepoint | N | MD | SD | ES | 95% CI | p |
|---|---|---|---|---|---|---|---|---|
| **HRQL** | EQ-5D-5L utility index | 5 months | 1217 | 0.127 | 0.201 | **0.630** | 0.569, 0.692 | <.001 |
| | | 7 months | 1050 | 0.137 | 0.204 | **0.669** | 0.602, 0.736 | <.001 |
| | | 9 months | 902 | 0.134 | 0.208 | **0.646** | 0.574, 0.718 | <.001 |
| | | 12 months | 778 | 0.142 | 0.226 | **0.631** | 0.554, 0.708 | <.001 |
| | QLQ-C30 summary score | 5 months | 1205 | 12.06 | 14.63 | **0.824** | 0.759, 0.890 | <.001 |
| | | 7 months | 1044 | 12.41 | 14.80 | **0.839** | 0.768, 0.909 | <.001 |
| | | 9 months | 898 | 12.47 | 15.32 | **0.814** | 0.738, 0.889 | <.001 |
| | | 12 months | 773 | 13.51 | 15.93 | **0.848** | 0.766, 0.930 | <.001 |
| **Pain** | QLQ-C30 pain subscale | 5 months | 1206 | 17.27 | 26.34 | **0.656** | 0.593, 0.718 | <.001 |
| | | 7 months | 1047 | 17.78 | 26.40 | **0.673** | 0.606, 0.740 | <.001 |
| | | 9 months | 899 | 17.52 | 27.59 | **0.635** | 0.563, 0.706 | <.001 |
| | | 12 months | 775 | 19.66 | 28.37 | **0.693** | 0.614, 0.771 | <.001 |
| **Sleep** | PROMIS sleep disturbance 8b | 5 months | 1195 | 6.890 | 9.358 | **0.736** | 0.672, 0.800 | <.001 |
| | | 7 months | 1037 | 6.862 | 9.464 | **0.725** | 0.656, 0.793 | <.001 |
| | | 9 months | 891 | 6.922 | 9.348 | **0.741** | 0.666, 0.814 | <.001 |
| | | 12 months | 768 | 7.828 | 10.06 | **0.778** | 0.697, 0.859 | <.001 |
| **Fatigue** | PROMIS fatigue 13a | 5 months | 1198 | 5.415 | 8.298 | **0.653** | 0.590, 0.715 | <.001 |
| | | 7 months | 1037 | 5.685 | 8.394 | **0.677** | 0.610, 0.745 | <.001 |
| | | 9 months | 893 | 5.685 | 8.756 | **0.649** | 0.577, 0.721 | <.001 |
| | | 12 months | 768 | 6.090 | 8.886 | **0.685** | 0.607, 0.764 | <.001 |
| **Depression** | DASS-21 depression subscale | 5 months | 1199 | 4.832 | 9.010 | **0.536** | 0.476, 0.597 | <.001 |
| | | 7 months | 1039 | 5.207 | 8.817 | **0.591** | 0.525, 0.656 | <.001 |
| | | 9 months | 895 | 5.377 | 8.697 | **0.618** | 0.526, 0.690 | <.001 |
| | | 12 months | 768 | 5.492 | 9.105 | **0.603** | 0.526, 0.680 | <.001 |
| **Anxiety** | DASS-21 anxiety subscale | 5 months | 1200 | 3.597 | 7.068 | **0.509** | 0.449, 0.569 | <.001 |
| | | 7 months | 1039 | 3.731 | 7.122 | **0.524** | 0.459, 0.589 | <.001 |
| | | 9 months | 895 | 3.423 | 6.988 | 0.490 | 0.420, 0.559 | <.001 |
| | | 12 months | 768 | 3.716 | 7.459 | 0.498 | 0.423, 0.573 | <.001 |

CI confidence interval of effect size; ES standardized mean-difference effect size (Cohen's d); HRQL health-related quality of Life; MD mean difference in direction of improvement; PROM patient-reported outcome measure; SD standard deviation of mean difference. p values reported are from paired t-tests. bold indicates clinically meaningful change (d ≥ 0.5).

### Pain.

QLQ-C30 pain subscale scores across the cohort showed significant linear ($t_{(9002)} = 10.60, p < 0.001$) and quadratic ($t_{(13012)} = 8.90, p < 0.001$) trends of improvement over time (Fig 4a). Mean scores improved by 14.39 (SD = 28.99; 95%CI: 14.19, 15.55; d = 0.5) with significantly greater improvements observed over time in participants with a chronic pain diagnosis (n = 1615) compared to those without ($t_{(5144)} = 11.17, p < 0.001$)(Fig 4b). Following guidelines, change of more than 14 points is considered a large clinical improvement [43].

**Pain intensity.** PROMIS Pain Intensity 3a scores showed significant linear ($t_{(1640)} = 15.39, p < 0.001$) and quadratic ($t_{(4072)} = 11.74, p < 0.001$) trends of improvement over time for participants with chronic pain (Fig 5a). Mean improvement in pain intensity T-scores from baseline to follow-up was 4.94 (SD = 6.53; 95%CI: 4.62, 5.26) indicating clinically meaningful improvement (d = 0.76) greater than the recommended PROMIS MCID of 3 T-scores.

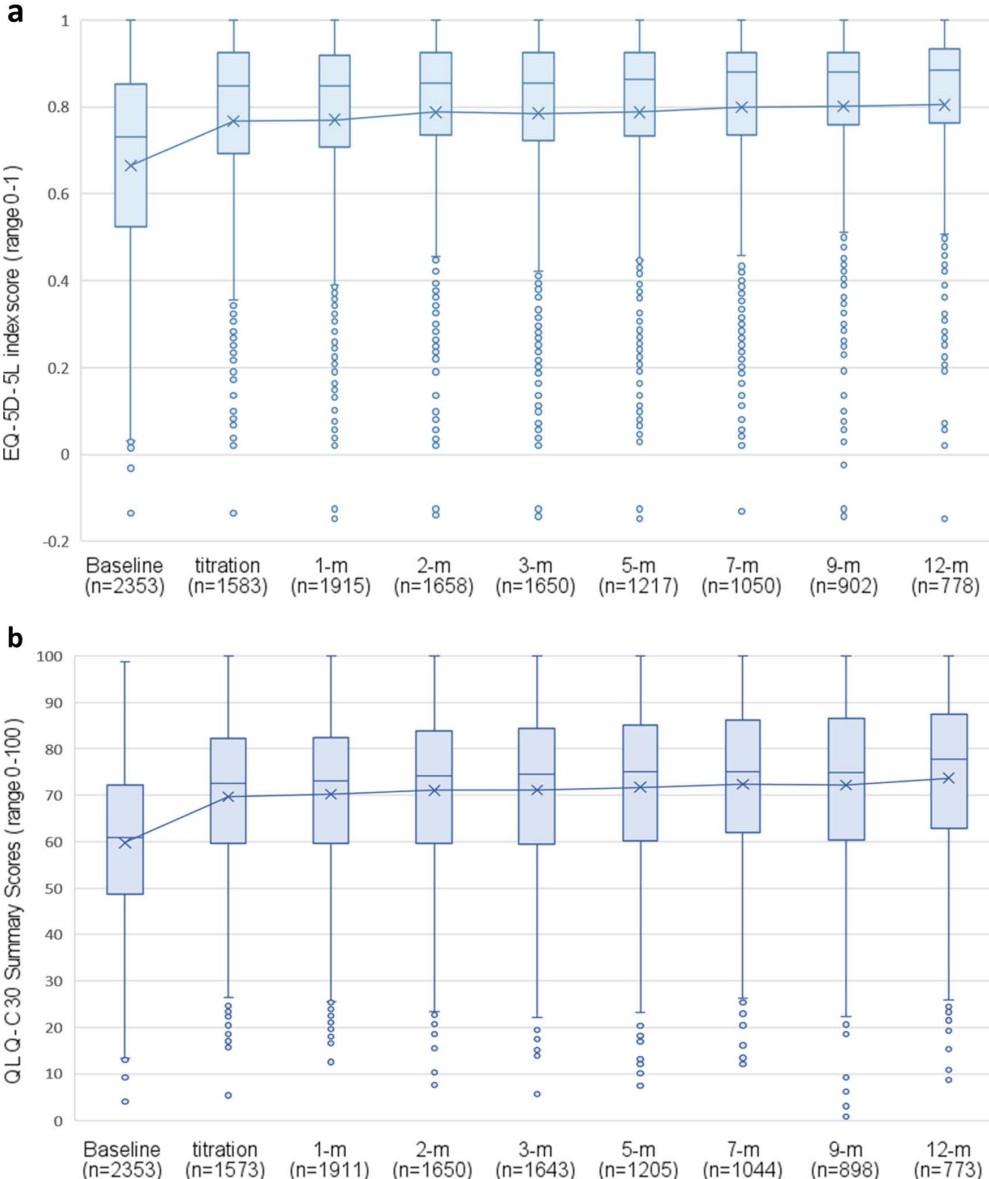

**Fig 3. Score distribution from baseline to 12-months following titration box plots with median bars and mean line for a) EQ-5D-5L Australian weighted Index Scores, and b) QLQ-C30 Summary Scores.** Higher scores indicate better quality of life.

When comparing change scores from baseline to follow-up across the different pain conditions (visceral, headache, musculoskeletal, neuropathic, cancer-related, or post-surgery), significant differences were observed between neuropathic and musculoskeletal pain (p = 0.028), neuropathic and headache pain (p = 0.019), and between headache and widespread pain (p = 0.027)(Fig 5b).

**Pain interference.** PROMIS Pain Interference 8a scores also showed significant linear ($t_{(1651)} = 15.16$, $p < 0.001$) and quadratic ($t_{(3853)} = 12.07$, $p < 0.001$) trends of improvement over time for participants with chronic pain (Fig 6a). Mean improvement in pain interference T-scores from baseline to average follow-up was 4.87 (SD = 6.44; 95%CI: 4.56, 5.19) indicating clinically meaningful improvement (d = 0.76) greater than the recommended PROMIS MCID of 3 T-scores.

**Table 3. Change in self-reported HRQL from baseline to mean post-therapy scores for participants exclusively treated for each health condition.**

| PROM | Health condition^ | N | MD | SD | ES | 95% CI | p |
|---|---|---|---|---|---|---|---|
| EQ-5D-5L utility index | | | | | | | |
| | Chronic pain† | 1024 | 0.118 | 0.186 | **0.64** | 0.57, 0.70 | <.001 |
| | Sleep disorder | 93 | 0.057 | 0.108 | **0.53** | 0.31, 0.75 | <.001 |
| | Generalised anxiety disorder | 202 | 0.090 | 0.145 | **0.62** | 0.47, 0.77 | <.001 |
| | Movement disorder‡ | 15 | 0.004 | 0.156 | 0.03 | −0.48, 0.53 | 0.923 |
| | PTSD | 22 | 0.106 | 0.194 | **0.55** | 0.09, 0.99 | 0.018 |
| | Mixed anxiety and depression | 94 | 0.061 | 0.146 | 0.42 | 0.21, 0.63 | <.001 |
| | Epilepsy | 10 | 0.000 | 0.091 | 0.00 | −0.62, 0.62 | 0.99 |
| QLQ-C30 summary score | | | | | | | |
| | Chronic pain† | 1022 | 9.559 | 11.88 | **0.81** | 0.73, 0.88 | <.001 |
| | Sleep disorder | 93 | 10.22 | 9.653 | **1.06** | 0.80, 1.31 | <.001 |
| | Generalised anxiety disorder | 200 | 10.68 | 10.79 | **0.99** | 0.82, 1.16 | <.001 |
| | Movement disorder‡ | 15 | 5.204 | 11.90 | 0.45 | −0.09, 0.97 | 0.106 |
| | PTSD | 19 | 12.85 | 10.86 | **1.18** | 0.58, 1.77 | <.001 |
| | Mixed anxiety and depression | 93 | 8.304 | 13.212 | **0.63** | 0.41, 0.85 | <.001 |
| | Epilepsy | 10 | 6.28 | 8.694 | **0.72** | 0.01, 1.35 | 0.048 |

ES: standardized mean-difference effect size (Cohen's d), **bold** indicates clinically meaningful change (d ≥ 0.5)

Higher scores indicate better health related quality of life (HRQL)

PTSD post-traumatic stress disorder

^ participants exclusively treated for the health condition listed

†Chronic pain conditions include neuropathic, widespread (fibromyalgia), primary and secondary musculoskeletal, primary and secondary headache or orofacial, primary and secondary visceral, cancer-related, and post-traumatic.

‡movement disorders included: Parkinsonism, tremor, paroxysmal dyskinesias, dystonia, ataxia, and tic disorders.

When comparing change scores from baseline to follow-up across the different pain conditions, significant differences were observed between visceral and neuropathic pain (p < 0.001), visceral and widespread pain (p = 0.008), visceral and musculoskeletal pain (p = 0.001), and between headache and neuropathic pain (p = 0.002), headache and widespread pain (p = 0.014), and headache and musculoskeletal pain (p = 0.002) (Fig 6b).

Table 4 reports mean difference of pain interference and pain intensity T-scores and effect size from baseline to each follow-up timepoint for participants with a pain condition.

## Sleep

PROMIS Sleep Disturbance T-scores showed a significant linear ($t_{(2546)} = 12.18$, p < 0.001) and quadratic ($t_{(4877)} = 16.96$, p < 0.001) trends of initial large improvement that was maintained over 12-months for the whole cohort(Fig 7a). Adjusted mean baseline scores (T = 61.35; SD = 8.61) improved by 5.96 points (SD = 7.81; 95%CI: 5.64, 6.27) to mean follow-up (T = 55.38; SD = 9.59), indicating clinically meaningful improvement greater than the recommended PROMIS MCID of 3 T-scores (d = 0.76), with participants having diagnosed insomnia improving significantly more than those without ($t_{(4806)} = 6.831$, p < 0.001)(Fig 7b).

Analysis of 546 participants with an insomnia diagnosis revealed statistically significant and clinically meaningful improvements in sleep disturbance of 7.96 (SD = 7.83; 95%CI: 7.30, 8.62; p < 0.001) from baseline (T = 63.91; SD = 7.42) to mean follow-up (T = 55.95; SD = 9.64), well above the recommended MCID (d = 1.02). Mean difference and effect size in sleep disturbance T-scores from baseline to each follow-up timepoint for participants with insomnia are reported in Table 4.

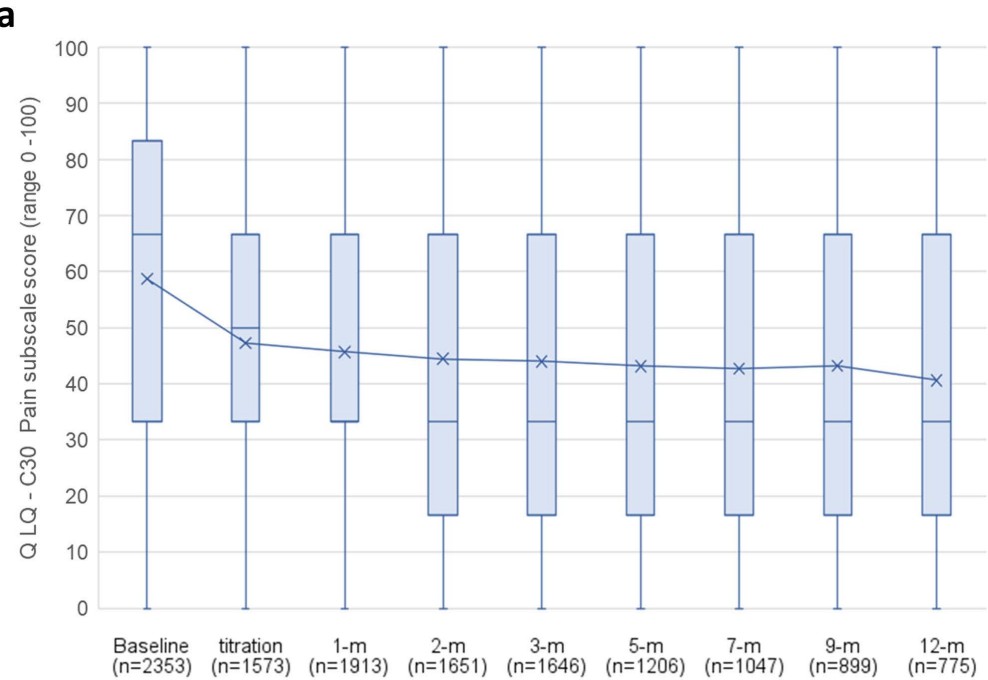

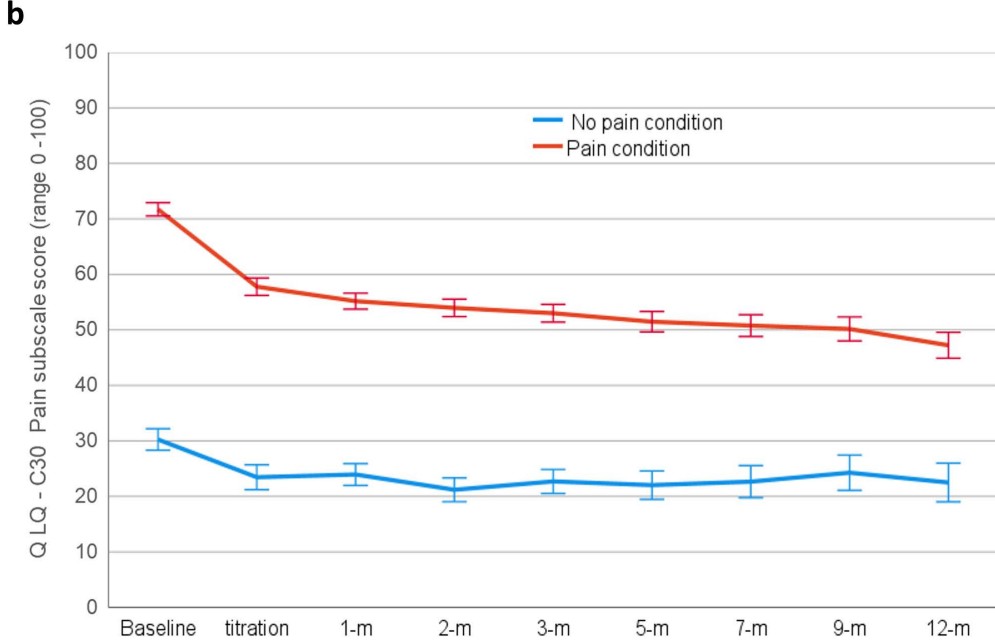

**Fig 4. QLQ-C30 Pain subscale scores from baseline to 12-months following titration for a) Score distribution box plot with median bars and mean line for whole cohort and b) Comparisons of mean scores for participants with a pain diagnosis vs no pain diagnosis.** Higher scores indicate greater symptom burden. Error bars are 95%CI.

## Fatigue

PROMIS Fatigue T-scores (n = 2353) displayed significant linear ($t_{(9430)} = 9.732, p < 0.001$) and quadratic ($t_{(12865)} = 9.437, p < 0.001$) trends of improvement over time (Fig 8). After adjustments, fatigue improved on average by 4.70 T-scores (SD = 9.25; 95%CI: 4.32, 5.07; p < 0.001) from baseline

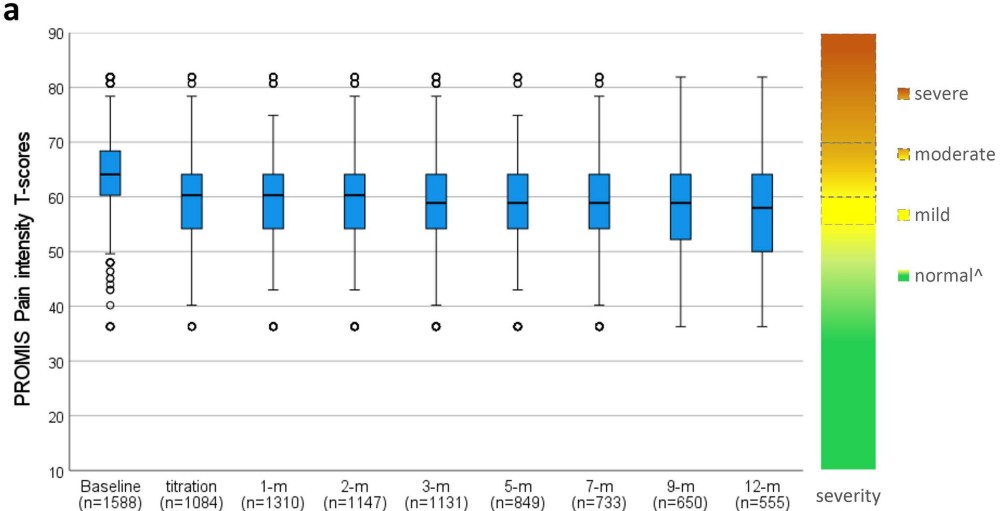

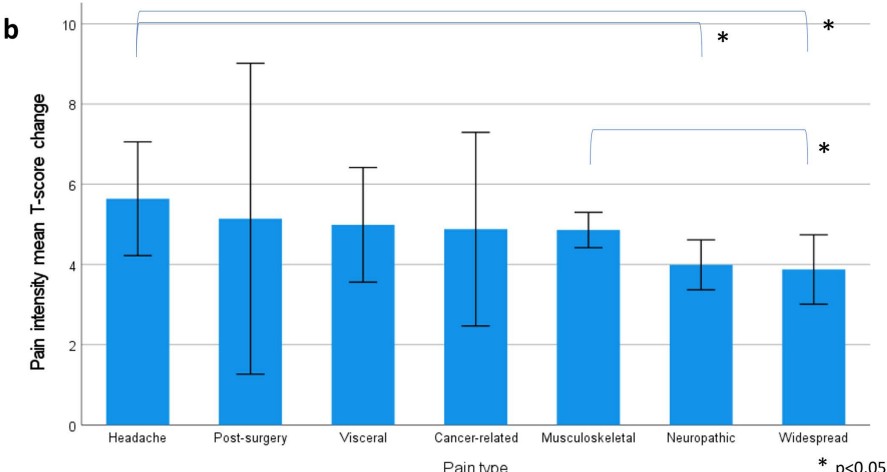

**Fig 5. PROMIS T-scores for participants with a pain diagnosis for a) Pain intensity T-score distribution from baseline to 12-months following titration with PROMIS severity scale b) Pain intensity mean difference from baseline to average follow-up across different pain conditions.**

(T = 60.85; SD = 9.01) to mean follow-up (T = 56.15; SD = 9.71), indicating clinically meaningful improvement greater than the recommended PROMIS MCID of 3 T-scores (d = 0.51).

## Depression

Mean DASS-Depression Scores displayed significant linear ($t_{(9826)} = 9.584, p < 0.001$) and quadratic ($t_{(12739)} = 7.674, p < 0.001$) trends of improvement for the whole cohort over time (Fig 9a). Mean difference between baseline (17.68; SD = 10.91) and average follow-up (10.73; SD = 10.18) was 4.53 (SD = 9.85; 95%CI: 4.13, 4.92; p < 0.001), not satisfying the recommended 5-point threshold for clinically meaningful improvement (d = 0.46). When comparing those with a depressive disorder to those without, improvements from baseline to mean follow-up were greater for the depression group ($t_{(11546)} = 4.852, p < 0.001$)(Fig 10a). After categorizing mean depression scores at each follow-up timepoint by DASS-recommended severity ratings (Fig

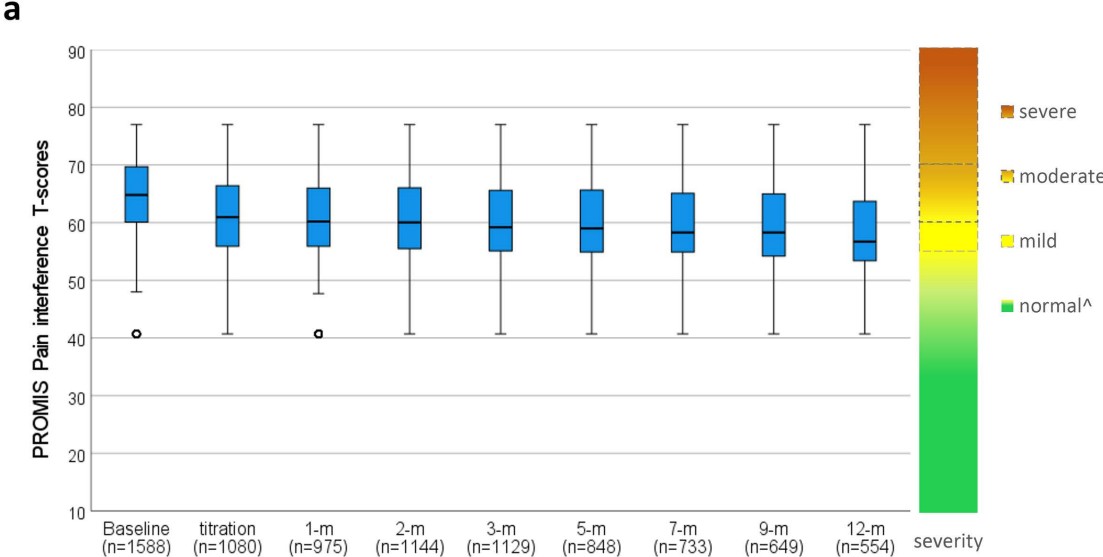

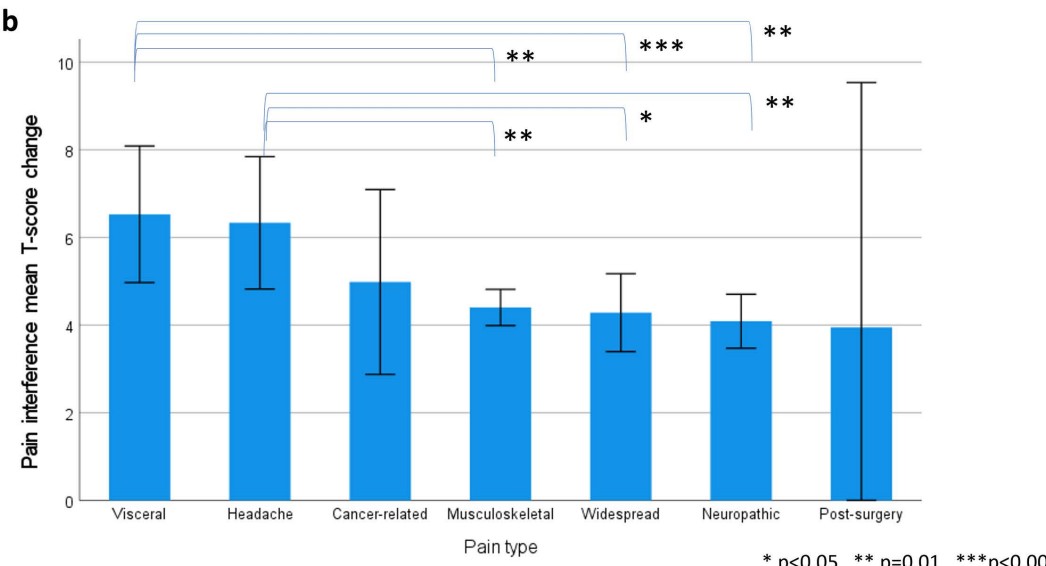

**Fig 6. PROMIS T-scores for participants with a pain diagnosis for a) Pain interference T-score distribution from baseline to 12-months following titration with severity scale, and b) Pain interference mean difference from baseline to average follow-up across different pain conditions.**

11a), mean follow-up category distribution compared with baseline demonstrated significant movement from more severe categories towards the normal range ($X^2 = 393$; df = 4; p < 0.001).

Examining 296 participants with a depressive disorder (i.e., mixed anxiety and depression, recurrent depressive disorder, or bipolar disorder), mean improvement in depression scores from baseline (22.55; SD = 11.07) to mean follow-up (14.06; SD = 11.23) was 7.19 (SD = 11.03; 95%CI: 5.93, 8.45; p < 0.001), demonstrating clinically meaningful improvement greater than 5 points and movement from severe range to moderate (d = 0.65). Mean difference and effect size of DASS-depression scores from baseline to each follow-up timepoint for participants with depressive conditions are reported in Table 4.

**Table 4. Mean difference in condition-specific PROM scores from baseline to each follow-up for participants receiving MC for those conditions.**

| Condition | PROM | Follow-up | N | MD | SD | ES | 95% CI | p |
|---|---|---|---|---|---|---|---|---|
| **Chronic** | PROMIS pain intensity 3a | Titration | 1084 | 3.80 | 6.32 | **0.60** | 0.54, 0.67 | <.001 |
| **Pain** | | 1 month | 1310 | 4.49 | 7.24 | **0.62** | 0.56, 0.68 | <.001 |
| (n = 1588) | | 2 months | 1147 | 5.02 | 7.59 | **0.66** | 0.60, 0.73 | <.001 |
| | | 3 months | 1131 | 5.39 | 7.50 | **0.72** | 0.65, 0.78 | <.001 |
| | | 5 months | 849 | 5.59 | 8.31 | **0.67** | 0.60, 0.75 | <.001 |
| | | 7 months | 733 | 5.50 | 8.07 | **0.68** | 0.60, 0.76 | <.001 |
| | | 9 months | 650 | 5.94 | 8.30 | **0.72** | 0.63, 0.80 | <.001 |
| | | 12 months | 555 | 6.57 | 8.47 | **0.77** | 0.68, 0.87 | <.001 |
| | PROMIS pain interference 8a | Titration | 1080 | 3.66 | 6.19 | **0.59** | 0.59, 0.66 | <.001 |
| | | 1 month | 975 | 4.42 | 7.07 | **0.63** | 0.56, 0.69 | <.001 |
| | | 2 months | 1144 | 5.00 | 7.58 | **0.66** | 0.60, 0.72 | <.001 |
| | | 3 months | 1129 | 5.39 | 7.46 | **0.72** | 0.66, 0.79 | <.001 |
| | | 5 months | 848 | 5.55 | 7.81 | **0.71** | 0.64, 0.79 | <.001 |
| | | 7 months | 733 | 5.61 | 8.00 | **0.70** | 0.62, 0.78 | <.001 |
| | | 9 months | 649 | 5.84 | 8.09 | **0.72** | 0.64, 0.81 | <.001 |
| | | 12 months | 554 | 6.49 | 8.23 | **0.79** | 0.69, 0.88 | <.001 |
| **Anxiety** | DASS-anxiety subscale | Titration | 511 | 4.95 | 7.00 | **0.71** | 0.61, 0.80 | <.001 |
| **Disorders** | | 1 month | 608 | 5.69 | 7.28 | **0.78** | 0.69, 0.87 | <.001 |
| (n = 775) | | 2 months | 501 | 5.84 | 7.76 | **0.75** | 0.65, 0.85 | <.001 |
| | | 3 months | 504 | 5.99 | 8.03 | **0.75** | 0.65, 0.84 | <.001 |
| | | 5 months | 349 | 5.91 | 7.83 | **0.76** | 0.64, 0.87 | <.001 |
| | | 7 months | 307 | 6.24 | 7.52 | **0.83** | 0.70, 0.96 | <.001 |
| | | 9 months | 254 | 6.21 | 7.80 | **0.80** | 0.65, 0.94 | <.001 |
| | | 12 months | 216 | 6.53 | 8.17 | **0.80** | 0.64, 0.95 | <.001 |
| **Depressive** | DASS-depression subscale | Titration | 195 | 6.50 | 9.60 | **0.68** | 0.52, 0.83 | <.001 |
| **Disorders** | | 1 month | 239 | 7.07 | 9.11 | **0.78** | 0.63, 0.92 | <.001 |
| (n = 296) | | 2 months | 203 | 6.51 | 9.18 | **0.71** | 0.55, 0.86 | <.001 |
| | | 3 months | 202 | 7.06 | 9.88 | **0.71** | 0.56, 0.87 | <.001 |
| | | 5 months | 143 | 7.83 | 11.13 | **0.70** | 0.52, 0.89 | <.001 |
| | | 7 months | 136 | 8.54 | 9.61 | **0.89** | 0.69, 1.09 | <.001 |
| | | 9 months | 116 | 8.34 | 9.07 | **0.92** | 0.70, 1.14 | <.001 |
| | | 12 months | 103 | 8.85 | 9.77 | **0.91** | 0.68, 1.13 | <.001 |
| **Insomnia** | PROMIS sleep disturbance 8b | Titration | 382 | 7.89 | 8.34 | **0.95** | 0.82, 1.07 | <.001 |
| (n = 546) | | 1 month | 455 | 7.82 | 8.54 | **0.92** | 0.81, 1.03 | <.001 |
| | | 2 months | 389 | 8.99 | 9.00 | **1.00** | 0.88, 1.12 | <.001 |
| | | 3 months | 379 | 9.23 | 9.35 | **1.00** | 0.86, 1.11 | <.001 |
| | | 5 months | 287 | 8.94 | 8.63 | **1.04** | 0.89, 1.18 | <.001 |
| | | 7 months | 246 | 8.93 | 9.00 | **0.99** | 0.84, 1.14 | <.001 |
| | | 9 months | 204 | 9.14 | 8.58 | **1.07** | 0.89, 1.24 | <.001 |
| | | 12 months | 183 | 9.67 | 9.34 | **1.04** | 0.86, 1.21 | <.001 |
| **Movement** | Neuro-QoL Upper function^ | Titration | 22 | 1.23 | 4.91 | 0.25 | 0.67, 0.18 | 0.252 |
| **Disorders** | | 1 month | 41 | 2.00 | 6.20 | 0.32 | 0.63, 0.01 | 0.046 |
| (n = 49) | | 2 months | 35 | 1.95 | 4.51 | 0.43 | 0.78, 0.08 | 0.015 |
| | | 3 months | 41 | 1.55 | 5.50 | 0.28 | 0.59, 0.03 | 0.079 |
| | | 5 months | 30 | 1.30 | 6.00 | 0.22 | 0.58, 0.15 | 0.246 |
| | | 7 months | 10 | 0.02 | 3.32 | 0.01 | 0.63, 0.61 | 0.985 |

*(Continued)*

**Table 4.** (Continued)

| Condition | PROM | Follow-up | N | MD | SD | ES | 95% CI | p |
|---|---|---|---|---|---|---|---|---|
| | | 9 months | 28 | 1.29 | 6.78 | 0.19 | 0.56, 0.18 | 0.322 |
| | | 12 months | 24 | 1.72 | 6.46 | 0.27 | 0.67, 0.14 | 0.206 |

ES standardized mean-difference effect size (Cohen's d); CI confidence interval of effect size; MC medicinal cannabis; MD mean difference in direction of improvement; SD standard deviation of mean difference; PROM patient-reported outcome measure

**bold** indicates clinically meaningful change (d ≥ 0.5 with 95%CI above 0.5)

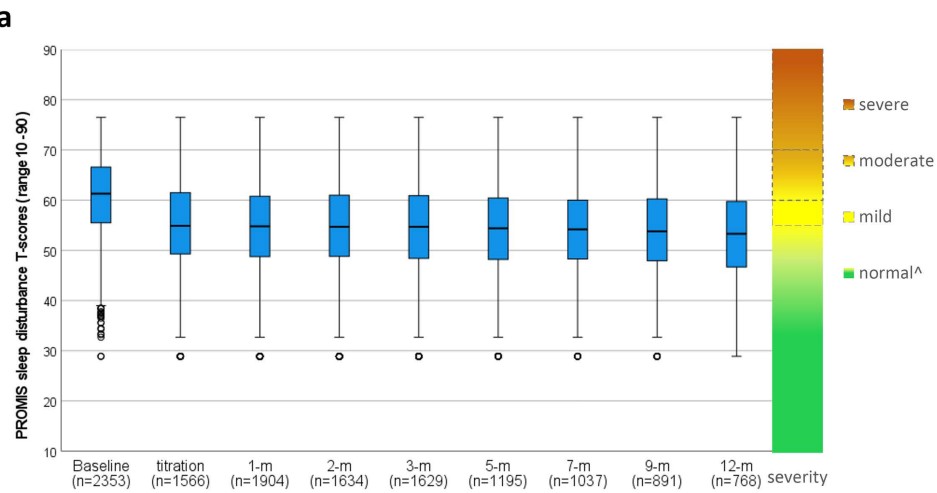

^50 is the mean for a sample representative of individuals with chronic conditions (not general population)

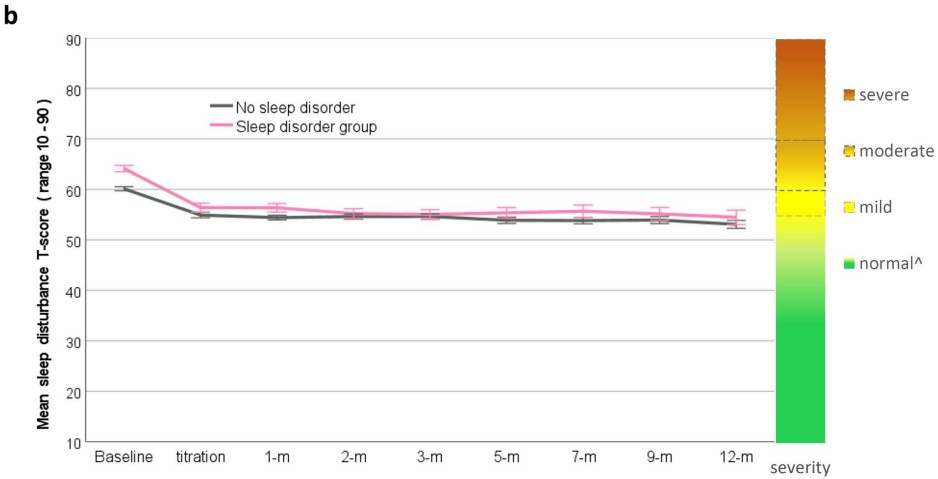

^50 is the mean for a sample representative of individuals with chronic conditions (not general population)

**Fig 7. PROMIS sleep disturbance T-scores from baseline to 12-months following titration with severity scale a) score distribution box plots for whole cohort, and b) Comparisons of mean scores for participants with a sleep disorder vs no sleep disorder.**

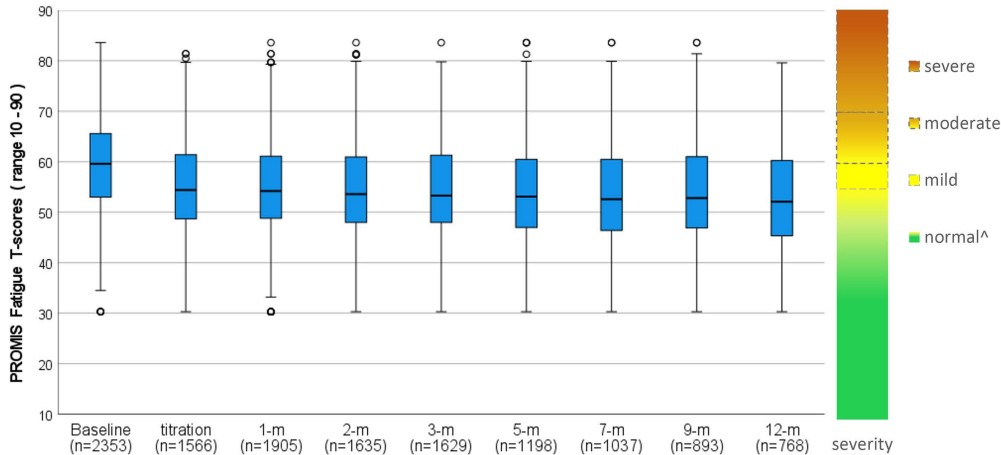

^50 is the mean for a sample representative of the general population

**Fig 8. PROMIS fatigue T-score distribution box plots from baseline to 12-months following titration for whole cohort.**

## Anxiety

Mean DASS-Anxiety scores displayed significant linear ($t_{(9641)} = 8.108, p < 0.001$) and quadratic ($t_{(12756)} = 8.360, p < 0.001$) trends of improvement over time (Fig 9b). Mean difference between baseline (11.85; SD = 8.80) and mean follow-up (8.58; SD = 7.48) was 3.27 (SD = 7.49; 95%CI:2.98, 3.57; p < 0.001), not reaching the recommended 5-point MCID threshold (d = 0.44). Comparing participants with anxiety conditions to those without anxiety, the improvement in DASS-anxiety scores from baseline to mean follow-up was greater for the anxiety group ($t_{(11383)} = 10.81$, p < 0.001) (Fig 10b). After categorizing anxiety scores at each timepoint by severity (Fig 11b), the average of follow-up distribution was compared with baseline showing significant change from more severe anxiety categories towards the normal range ($X^2 = 372$; df = 4; p < 0.001).

Examining the 775 participants with anxiety health conditions (i.e., generalised anxiety or mixed depression and anxiety), the mean change from 15.44 (SD = 8.93) at baseline to mean follow-up 9.79 (SD = 8.17) was 5.65 (SD = 8.21; 95%CI: 5.07, 6.23; p < 0.001), indicating a clinically meaningful improvement with change larger than 5 points from severe category to moderate (d = 0.69). Mean difference and effect size of DASS-anxiety scores from baseline to each follow-up timepoint for participants with anxiety conditions are displayed in Table 4.

## Movement disorder

After adjusting for age, sex, and pain duration, there were no significant linear or quadratic trends of change over time in Neuro-QoL Adult Upper Extremity Function scores (p-values 0.58 and 0.42 respectively) among participants diagnosed with movement disorder (n = 49). Compared to baseline, average follow-up T-scores improved by 1.43 (SD = 6.65; 95%CI: -0.44, 3.30; p = 0.134), not meeting the recommended MCID of 3 T-scores (d = 0.21). Mean difference and effect size of Neuro-QoL Upper Extremity Function T-scores from baseline to each follow-up timepoint for participants with movement disorders are reported in Table 4, showing no clinically meaningful changes at any follow-up timepoint.

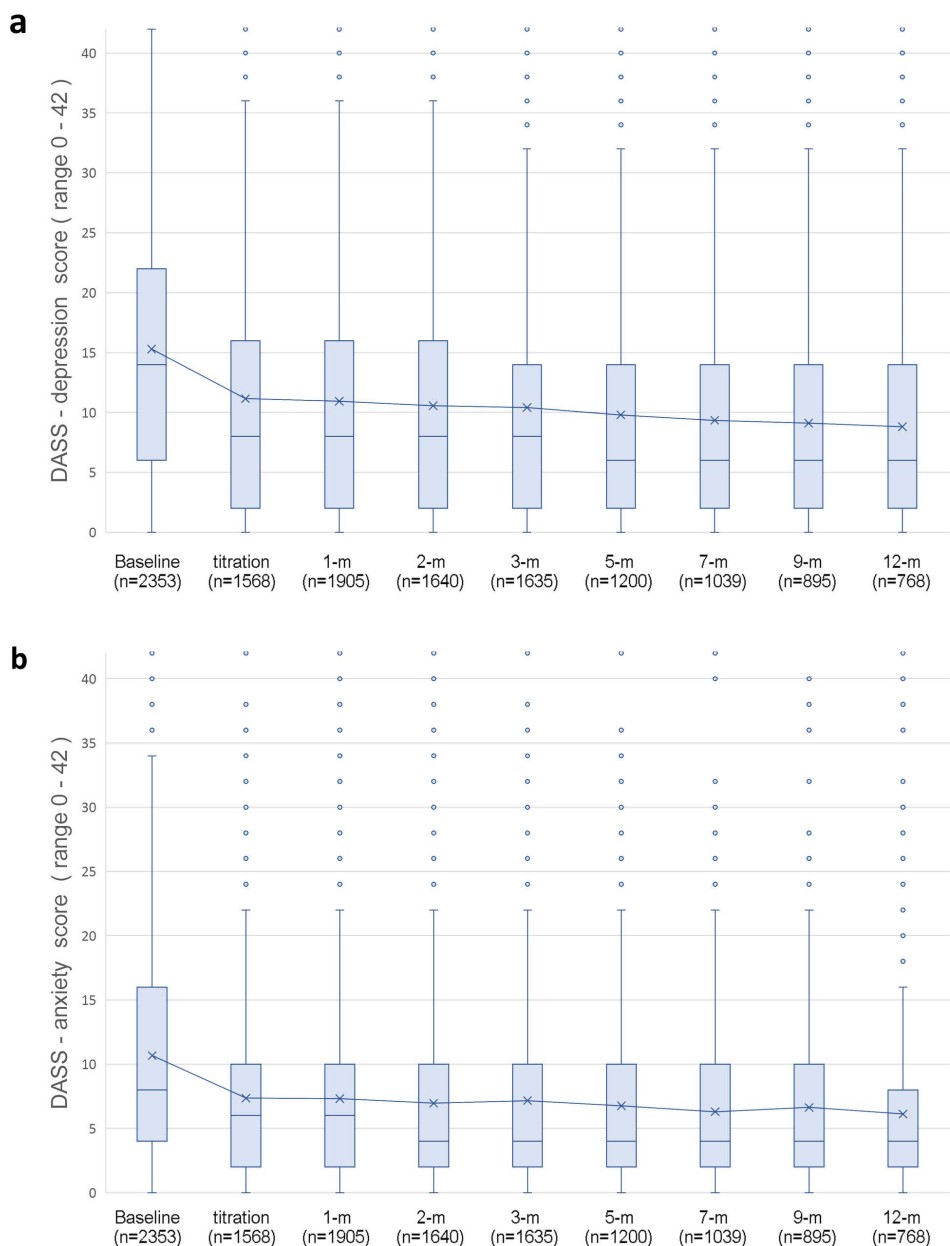

**Fig 9. Mean DASS score distribution box plots with median bars and mean line from baseline to 12-months following titration across the whole cohort for a) depression subscale and b) anxiety subscale.** Higher scores indicate greater symptom burden.

## Medicinal cannabis

When exploring the four MC composition categories, differences were observed in the degree of improvement in anxiety, depression, sleep, and fatigue across all participants. In all cases, average daily doses that were THC-dominant had greater odds of larger improvements in these outcomes compared with THC:CBD-balanced (Table 5), although CBD-dominant was also better than THC:CBD-balanced for anxiety. No differences in degree of improvement in pain intensity and interference were observed between MC compositions when looking at chronic pain patients as a group.

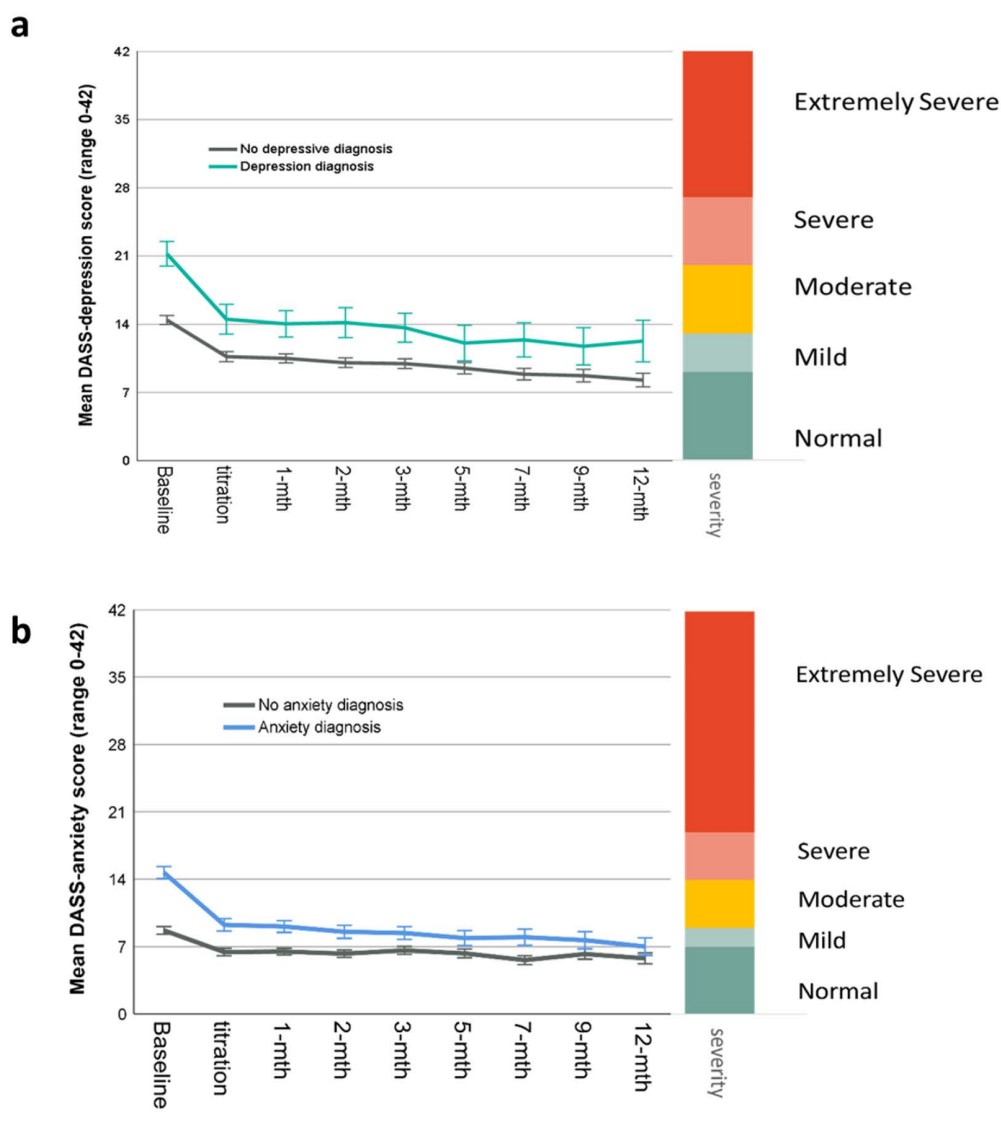

**Fig 10. Mean DASS scores at each timepoint plotted with DASS condition-specific severity scales for a) depression subscale for participants with a diagnosed depressive condition vs those without depression and b) anxiety subscale showing participants with a diagnosed anxiety condition vs those without anxiety.** Error bars are 95%CI.

On further exploration of different pain types, we found that CBD-dominant daily doses were associated with a greater degree of improvement in pain intensity for: musculoskeletal pain compared with CBD-only (OR:1.57; 95%CI: 1.12, 2.2; p = 0.013); headache pain compared to CBD:THC-balanced (OR:4.0; 95%CI: 1.3, 12; p = 0.015); and cancer-related pain when compared to CBD:THC-balanced (OR:7.3; 95%CI: 1.0, 49; p = 0.047). Pain interference improvements did not differ by cannabinoid combination.

## Missed assessments

Analyses using linear mixed models included all available data from PROMs completed at each timepoint. No items within completed PROMs were missed. Fig 12 shows EQ-5D and QLQ-C30 results stratified by those who dropped out or failed to complete follow-up after each timepoint. Trajectories over time suggest that after titration,

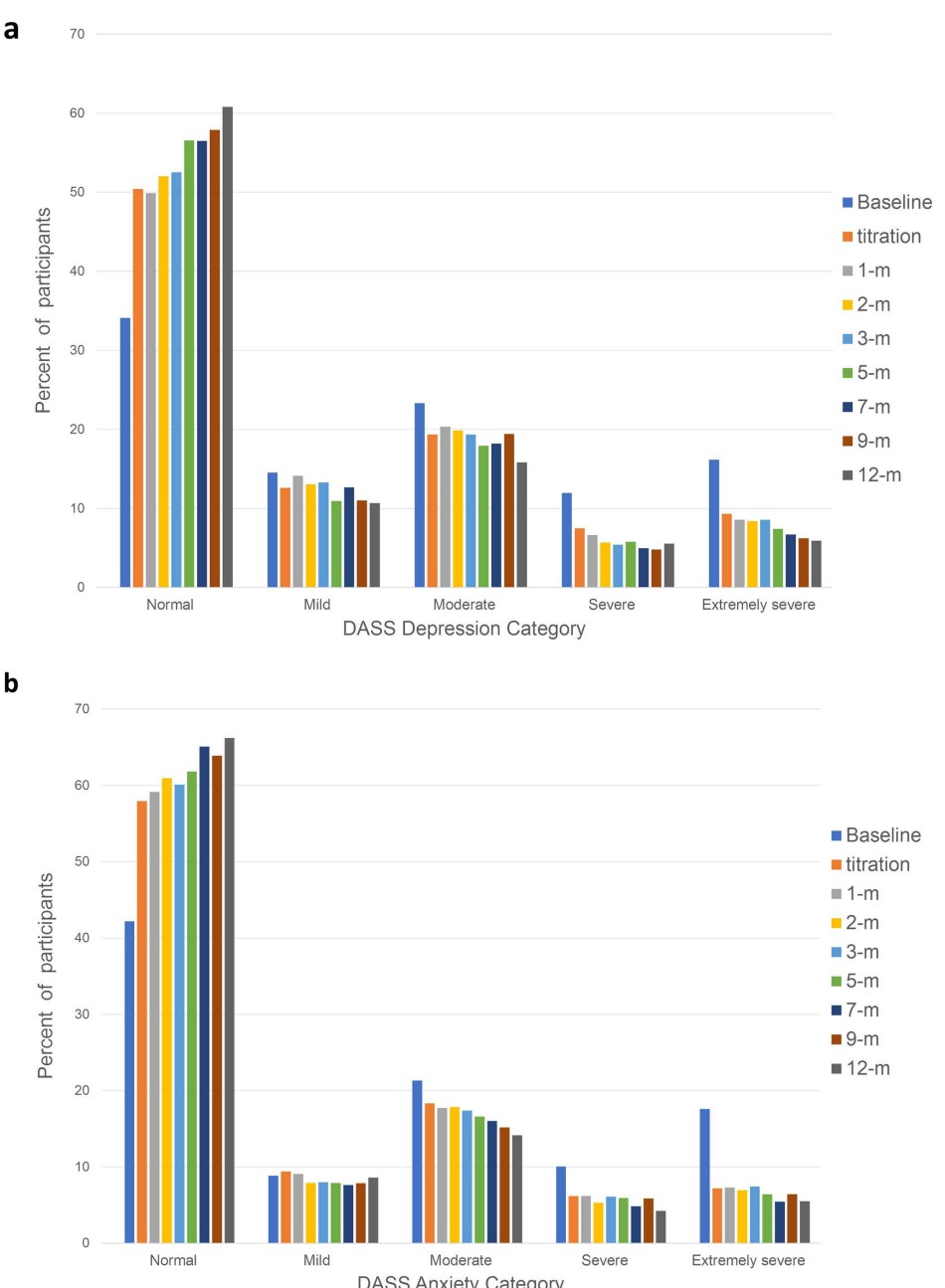

**Fig 11. Percent of participants with DASS scores falling within condition-specific severity categories at each timepoint for a) depression subscale scores and b) anxiety subscale scores.**

participants dropping out of the study at each timepoint had experienced a decline in HRQL since their previous assessments (with the only exception observed at 2-months). Participants who dropped out of the study before 3-months had better HRQL at baseline, and smaller improvements from baseline, than those who remained on the study for 3-months or more. There were no significant differences in HRQL scores observed between participants included in the study analyses and those who only completed baseline.

**Table 5. Mean change in PROM scores and odds of greater degree of improvement for each MC composition category.**

| Condition Group PROM | CBD-only M (SD) | CBD-dom. M (SD) | Balanced M (SD) | THC-dom. M (SD) | MC comparisons | OR (95%CI) ^ | p† |
|---|---|---|---|---|---|---|---|
| **All conditions, n** | 892 | 509 | 637 | 219 | | | |
| DASS - Anxiety | 3.17 (5.92) | 3.33 (6.26) | 2.6 (5.91) | 3.75 (6.21) | CBD-only – CBD-dom. | 0.95 (0.78, 1.16) | 0.624 |
| | | | | | CBD-only – Balanced | 1.19 (0.99, 1.43) | 0.069 |
| | | | | | CBD-only – THC-dom. | 0.84 (0.64, 1.1) | 0.198 |
| | | | | | **CBD-dom. – Balanced** | **1.24 (1.01, 1.53)** | **0.045** |
| | | | | | CBD-dom. – THC-dom. | 0.89 (0.67, 1.18) | 0.410 |
| | | | | | **THC-dom. – Balanced** | **1.41 (1.06, 1.85)** | **0.015** |
| DASS - Depression | 4.04 (7.17) | 4.47 (7.28) | 3.76 (7.40) | 5.12 (8.21) | CBD-only – CBD-dom. | 0.92 (0.76, 1.12) | 0.417 |
| | | | | | CBD-only – Balanced | 1.10 (0.92, 1.33) | 0.304 |
| | | | | | CBD-only – THC-dom. | 0.79 (0.60, 1.03) | 0.081 |
| | | | | | CBD-dom. – Balanced | 1.19 (0.96, 1.47) | 0.103 |
| | | | | | CBD-dom. – THC-dom. | 0.86 (0.64, 1.14) | 0.289 |
| | | | | | **THC-dom. – Balanced** | **1.39 (1.04, 1.82)** | **0.023** |
| Sleep disturbance | 4.67 (7.11) | 6.67 (8.13) | 6.27 (7.90) | 7.67 (7.81) | **CBD-dom. – CBD-only** | **1.61 (1.33, 1.96)** | **<0.001** |
| (PROMIS) | | | | | **Balanced – CBD-only** | **1.47 (1.22, 1.79)** | **<0.001** |
| | | | | | **THC-dom. – CBD-only** | **2.13 (1.61, 2.78)** | **<0.001** |
| | | | | | CBD-dom. – Balanced | 1.09 (0.89, 1.35) | 0.406 |
| | | | | | CBD-dom. – THC-dom. | 0.80 (0.60, 1.06) | 0.122 |
| | | | | | **THC-dom. – Balanced** | **1.39 (1.04, 1.82)** | **0.024** |
| Fatigue | 4.41 (6.61) | 4.83 (6.67) | 4.10 (6.48) | 5.71 (6.96) | CBD-only – CBD-dom. | 0.89 (0.73, 1.09) | 0.257 |
| (PROMIS) | | | | | CBD-only – Balanced | 1.09 (0.91, 1.31) | 0.359 |
| | | | | | **THC-dom. – CBD-only** | **1.43 (1.09, 1.85)** | **0.010** |
| | | | | | CBD-dom. – Balanced | 1.22 (0.99, 1.51) | 0.062 |
| | | | | | CBD-dom.– THC-dom. | 0.79 (0.59, 1.05) | 0.107 |
| | | | | | **THC-dom. – Balanced** | **1.56 (1.18, 2.04)** | **0.002** |
| **Chronic pain, n** | 538 | 362 | 450 | 167 | | | |
| Pain intensity | 4.48 (6.18) | 5.05 (6.81) | 4.35 (6.38) | 4.78 (6.18) | CBD-only – CBD-dom. | 0.85 (0.67, 1.09) | 0.194 |
| (PROMIS) | | | | | CBD-only – Balanced | 1.04 (0.83, 1.30) | 0.755 |
| | | | | | CBD-only – THC-dom. | 0.92 (0.67, 1.25) | 0.577 |
| | | | | | CBD-dom. – Balanced | 1.21 (0.94, 1.56) | 0.136 |
| | | | | | CBD-dom. – THC-dom. | 1.08 (0.77, 1.51) | 0.626 |
| | | | | | Balanced – THC-dom. | 0.88 (0.64, 1.22) | 0.417 |
| Pain interference | 4.59 (6.33) | 5.07 (6.24) | 4.23 (6.35) | 4.19 (4.83) | CBD-only – CBD-dom. | 0.87 (0.68, 1.11) | 0.258 |
| (PROMIS) | | | | | CBD-only – Balanced | 1.11 (0.88, 1.39) | 0.377 |
| | | | | | CBD-only – THC-dom. | 1.13 (0.82, 1.55) | 0.457 |
| | | | | | CBD-dom. – Balanced | 1.28 (0.99, 1.64) | 0.059 |
| | | | | | CBD-dom. – THC-dom. | 1.32 (0.94, 1.83) | 0.078 |
| | | | | | Balanced – THC-dom. | 1.01 (0.73, 1.40) | 0.944 |

CBD-only contains > 98% cannabidiol; CBD-dom. contains > 60% to 98% cannabidiol; Balanced contains 40% to 60% of both cannabidiol and delta5-tetrahydrocannabinol; THC-dom. contains > 60% to 98% delta5-tetrahydrocannabinol.

^Odds ratio effect size of mean difference in change scores.

†p values of independent samples T-tests.

**Bold** indicates significantly greater odds of improvement after Hochberg adjustment for multiple comparisons.

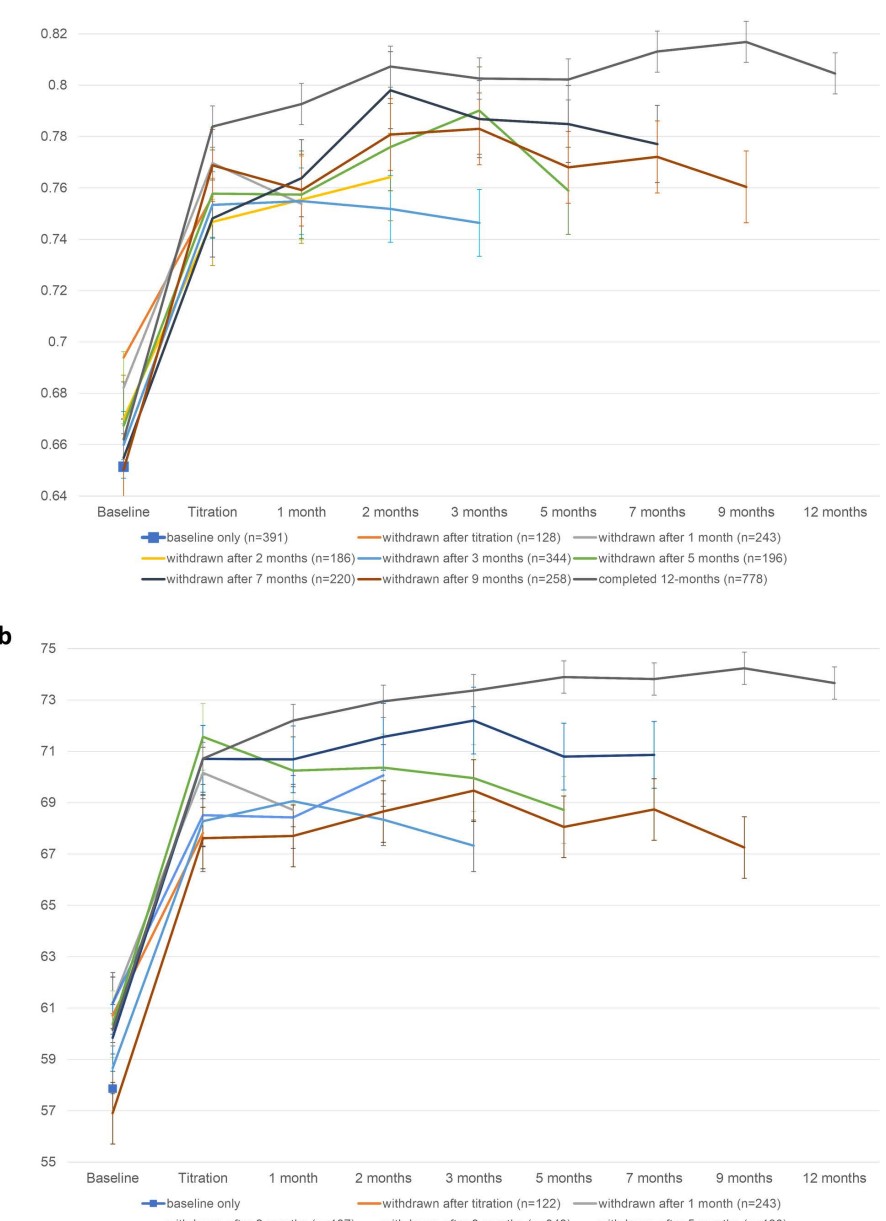

**Fig 12. Change in HRQL scores over 12-months stratified by time on study for a) mean EQ-5D-5L Utility Index scores and b) mean QLQ-C30 Summary scores.**

## Discussion

### Principal findings

We found that short term improvements in overall HRQL reported at 3-months[20] were maintained over a 12-month period in patients prescribed MC in Australia. Statistically significant and clinically meaningful improvements were observed in HRQL, fatigue, pain, and sleep for people with chronic health conditions. Similar improvements were found in pain

outcomes for participants with chronic pain; sleep disturbance for participants with insomnia; depression scores for patients with depression; and anxiety scores in patients with anxiety. Participants with movement disorders had improved HRQL but no significant improvements in upper extremity function scores.

## Comparison with other medicinal cannabis studies assessing PROs

HRQL improvements observed in our study are consistent with results published in 2023 from a UK registry of 312 patients with chronic health conditions prescribed MC reporting EQ-5D-5L index score improvements over 6-months (n = 63), and a 2022 Canadian registry following 2073 participants finding EQ-5D-5L health status improvements maintained up to 12-months (n = 600) [44]. HRQL improvements over 12-months were also observed in another cohort study of patients with chronic pain [45]. Our findings further revealed that HRQL improved in patients treated exclusively for non-pain conditions, such as insomnia, generalized anxiety, mixed depression and anxiety, and PTSD.

Similarly, Aviram et al. observed clinically meaningful improvements (greater than 30% change) in sleep disturbance, anxiety, depression, and affective pain from baseline to 12-months in an Israeli multicentre, prospective study of 551 patients with chronic pain receiving MC [46].

Clinically significant improvements in Brief Pain Inventory pain severity and pain interference scores were observed at 1-, 3-, 6-, and 12-months follow-up compared to baseline in an observational study of chronic pain patients, [45] which were similar to our findings using PROMIS pain intensity and interference PROMs.

An observational study by Safakish et al. found chronic pain patients (n = 248) experienced significant improvements in fatigue after 3-months of MC therapy,[45] and similar results have been reported for cancer patients (n = 743) [47].

A randomised crossover double-blind placebo-controlled trial with 29 adults with insomnia found that medicinal cannabis oil was effective in reducing insomnia severity index scores over a 2-week period, [48] and an observational study in 2021 reported Pittsburgh Sleep Quality Index score improvements after 3-months in 36 chronic pain patients prescribed MC [49]. In most previous studies examining sleep outcomes in patients treated with MC, validated PROMs were seldom used and many had small sample sizes or short treatment and follow-up periods [50]. Our results extend previous findings by indicating sleep improvements observed in insomnia patients treated with MC are maintained long-term. Similarly, results from an Australian registry of MC patients published in 2023 also showed significant improvements across the cohort in Insomnia Severity Index scores (n = 1902), Brief Pain Inventory severity and interference scores (n = 1651), and DASS anxiety and depression scores (n = 1874), after 12-months of MC therapy, which were maintained up to 2-years [51].

Sagar et al. published preliminary results of their longitudinal study in 2021, finding that participants with various health conditions treated with MC showed significant improvements from baseline to 6-months (n = 44) and 12-months (n = 32) in Pittsburgh Sleep Quality Index scores, Beck Depression Inventory scores, and in State-Trait Anxiety Inventory - trait anxiety scores [52]. Similar to our findings, an observational study by Rapin et al. reported that patients with baseline moderate to severe depression (n = 115) and anxiety (n = 138) showed clinically meaningful improvements in depression and anxiety scores respectively after 3-months that were maintained after 6-months of MC therapy[53].

When looking at opioids for pain management, most of our participants had reduced or stopped their opioid intake by the end of the study. Similar findings have also been previously reported by Pritchett et al., [54] where 79% of participants reduced or stopped opioids after

starting MC, and in a Canadian study that found the number of participants using opioids more than halved 6-months post MC therapy [55].

High quality evidence from randomized clinical trials (RCTs) suggested CBD reduces seizure frequency in epilepsy patients, however, in a real world setting there are concerns that unwanted side-effects may result from interactions with other anti-seizure medications [56]. This may explain the low numbers of participants with epilepsy recruited to this study. The small number of participants being treated for epilepsy exclusively (n = 10) may have led to the inconsistent findings observed in HRQL, with significant improvement in QLQ-C30 scores but no change in EQ-5D. Cannabis use in general has previously been associated with poorer HRQL and worse outcomes in adult epilepsy patients, [57] however this may not apply to MC with controlled dosing monitored by clinicians and requires further research with larger sample sizes

We observed differences in the degree of improvement in fatigue, insomnia, anxiety, and depression depending on the ratio of CBD and THC in average daily MC doses. For these outcomes, average daily doses of THC-dominant MC was associated with greater odds of improvement than CBD:THC-balanced MC. An Australian cross-sectional study by Trevitt et al. had similar findings regarding participants' self-rated global impression of change in anxiety when prescribed MC, however they also found improvements in pain for those prescribed THC-dominant products, and no differences regarding sleep [58]. In contrast, we observed that any daily MC doses containing THC were associated with a greater degree of improvement in sleep compared with CBD-only, and that overall, patients with chronic pain conditions did not report improvements in pain interference differently depending on CBD:THC composition. However, further exploration of the different pain types did reveal differences in musculoskeletal, headache, and cancer-related pain intensity, but this favored CBD-dominant doses. Participants on our study were taking lower daily doses averaging up to 50mg of CBD, compared with doses reported in RCTs that typically only find minimal improvements in anxiety, insomnia, and pain relief at much higher doses of CBD (300mg) [59]. However, RCTs often test the immediate (within hours), or short-term (in weeks), effects of CBD on outcomes and do not account for ongoing therapy over months or years. Our findings suggest that people with chronic pain conditions experience better outcomes over time on lower doses of CBD when combined with smaller amounts of THC at a ratio of 10:1. However, we calculated average daily dose overall, whereas daily dosing regimens in practice may limit administering THC to evenings (to avoid possible intoxication during waking hours), rather than maintaining a 10:1 ratio throughout the day. Our findings suggest there are clinically important differences in outcomes for some conditions depending on the ratio of THC and CBD, however as an observational study, we cannot infer efficacy of different dosage regimens conclusively. As MC becomes increasingly used and accepted globally, further research with multicentre RCTs is needed in this area to provide clinicians with evidence-based guidance on condition-specific MC prescribing and the efficacy of various dosing regimens for their patients.

## Strengths and limitations

Our study was large enough to assess patients across a wide range of chronic conditions and socio-demographics in a real-world setting. We recruited participants from more than 100 sites across Australia, covering most states and territories. We used validated, condition-relevant PROMs at clinically meaningful time-points allowing comparisons within groups over time, and between studies and patient groups. We reported the clinical meaningfulness of findings using predefined MCIDs, and determined MCIDs provided by PROM developers aligned with the Norman et al. MCID recommendation of half a standard deviation in

patients with chronic conditions [42]. By using a homogenous selection of MC oils (LGP products) our results were not confounded by the effects of different strains of cannabis plant or by different routes of administration. Unlike the large doses of CBD or THC typically administered in RCTs,[60] participants in our study were patients who titrated to doses used in clinical practice. The use of de-identified, automated, electronic data collection reduced the risk of response biases introduced when collecting identifiable PROM data in person.

However, as a single arm observational study, it is not possible to confidently attribute changes over time to the intervention. Within-group studies of cannabis and HRQL without control groups tend to report larger effect sizes than RCTs [61]. Our observed improvements may be due in part to a placebo effect, [62] mere-measurement effect, where PROs improve in response to completing PROMs, [63] or regression towards the mean, where scores fluctuate around a true mean and our observed scores maintained at subsequent timepoints are driven by those remaining on the study [64].

Although there were no significant differences in HRQL between participants who dropped out at baseline and those who remained on the study, the loss of participants at follow-up may have led to attrition bias. Participants remaining on the study were likely to be benefitting from MC, and those dropping out may have had reduced benefit considering the small decline in HRQL observed immediately before drop-out. Participants were asked to provide reasons for study withdrawal, however only 322 volunteered this information (132 due to lack of therapeutic benefit (41%)) and many did not respond. Despite standardizing the cost of MC for participants in the study, results may have been biased due to the financial burden of purchasing MC products not Government-subsidised in Australia. It is possible participants in the study were wealthier than typical patients with chronic conditions. Although 64 participants advised that they withdrew due to unwanted side-effects (20% of those who provided reasons), adverse events were not collected in this observational study. However, there were no reports of significant adverse effects to the product manufacturer. Lastly, participants were only prescribed LGP MC oil products which limits generalizability to other MC products and forms of administration (e.g., vapourised, tincture, patches). However, we were able to determine average daily doses of CBD and THC, which can be applied to other oil products.

### Clinical implications

In clinical practice, prescribing MC to patients with chronic health conditions may improve patients' pain, fatigue, insomnia, anxiety, and depression and overall HRQL. Current clinical guidelines support prescribing MC to patients who are interested in trialling it for conditions not responding to conventional treatments, [65] and our findings suggest any improvements would be apparent quickly and maintained long-term. Evidence on optimal CBD:THC ratios for different health conditions is emerging and will improve prescribing practices.

### Conclusion

Long-term findings over 12-months indicate patients prescribed MC in practice have improved HRQL and reduced fatigue. Patients with anxiety, depression, insomnia, or chronic pain diagnoses also improved over 12-months in condition-specific symptoms. We did not find conclusive evidence of motor function improvement in patients with movement disorders. Patients exclusively treated for generalized anxiety, chronic pain, insomnia, and PTSD, all showed improvements in HRQL. The findings from this study contribute to the emerging evidence-base to inform decision making both in clinical practice and at policy level.

## Supporting information

**S1 Table. Condition-specific outcomes assessed including characteristics, scoring, and details of use, for PROMs administered to QUEST participants with diagnosed chronic pain or movement disorder.**
(PDF)

**S2 Table. Patient-reported gender identity and ethnicity of 2744 participants recruited to the QUEST Initiative by those included in the 12-month analyses and those who completed baseline only.**
(PDF)

**S3 Table. Conditions treated with medicinal cannabis for 2744 participants recruited to the QUEST 12-month study.**
(PDF)

## Acknowledgements

The authors thank all the patients who participated in the QUEST study. We also thank Arthritis Australia, Epilepsy Action Australia, Health Insurance Fund Australia, and MS Research Australia for promoting the study, and the clinicians who identified patients eligible to receive study invitations, including: Dr Feroz Ameerjan, Dr Anthony Balint, Dr Mahala Buckley, Dr Alex Burmey, Dr Ceinwen Carlsson, Dr Vivienne Cebola, Dr David Corbet, Dr Michael Corbett, Dr Natasha Feingold, Dr David Gaskell, Dr Igor Jakubowicz, Dr Joe Kosterich, Dr Liling Leow, Dr Bentley Logan, Olga Lutzko NP, Dr Deb Mills, Dr Nancy Momoff, Dr Matty Moore, Simone O'Brien NP, Dr Bara Qattan, Dr Yan Ren, Dr Jamie Rickcord, Dr Stephan Rudzki, Dr Abdulmuminu Sambo, Dr James Stewart, Dr Joel Wren, and Dr Su-Yin Yeong.

## Author contributions

**Conceptualization:** Margaret-Ann Tait, Daniel SJ Costa, Rachel Campbell, Leon N Warne, Richard Norman, Stephan Schug, Claudia Rutherford.

**Data curation:** Margaret-Ann Tait, Claudia Rutherford.

**Formal analysis:** Margaret-Ann Tait, Daniel SJ Costa, Richard Norman, Claudia Rutherford.

**Funding acquisition:** Margaret-Ann Tait, Leon N Warne, Claudia Rutherford.

**Investigation:** Margaret-Ann Tait, Daniel SJ Costa, Leon N Warne, Richard Norman, Stephan Schug, Claudia Rutherford.

**Methodology:** Margaret-Ann Tait, Daniel SJ Costa, Rachel Campbell, Richard Norman, Stephan Schug, Claudia Rutherford.

**Project administration:** Margaret-Ann Tait, Leon N Warne.

**Resources:** Margaret-Ann Tait, Rachel Campbell, Leon N Warne, Richard Norman, Claudia Rutherford.

**Software:** Margaret-Ann Tait, Leon N Warne.

**Supervision:** Daniel SJ Costa, Claudia Rutherford.

**Validation:** Margaret-Ann Tait, Daniel SJ Costa, Rachel Campbell, Richard Norman, Stephan Schug, Claudia Rutherford.

**Visualization:** Margaret-Ann Tait, Daniel SJ Costa, Rachel Campbell, Leon N Warne, Richard Norman, Stephan Schug, Claudia Rutherford.

**Writing – original draft:** Margaret-Ann Tait.

**Writing – review & editing:** Margaret-Ann Tait, Daniel SJ Costa, Rachel Campbell, Leon N Warne, Richard Norman, Stephan Schug, Claudia Rutherford.

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
