## [Decision Letter · Decision Letter 0]

28 Nov 2024

Dear Dr. Tait,

Thank you for submitting your manuscript to PLOS ONE. After careful consideration, we feel that it has merit but does not fully meet PLOS ONE’s publication criteria as it currently stands. Therefore, we invite you to submit a revised version of the manuscript that addresses the points raised during the review process.

We look forward to receiving your revised manuscript.

Kind regards,

Francesca Baratta, PharmD, PhD

Academic Editor

PLOS ONE

Reviewers' comments:

Reviewer's Responses to Questions

**Comments to the Author**

1. Is the manuscript technically sound, and do the data support the conclusions?

Reviewer #1: Yes

Reviewer #2: Partly

2. Has the statistical analysis been performed appropriately and rigorously?

Reviewer #1: Yes

Reviewer #2: Yes

3. Have the authors made all data underlying the findings in their manuscript fully available?

Reviewer #1: Yes

Reviewer #2: Yes

4. Is the manuscript presented in an intelligible fashion and written in standard English?

Reviewer #1: Yes

Reviewer #2: No

Reviewer #1: Thanks for inviting me to review this paper exploring patient reported outcomes following medical cannabis treatment. I have a few comments:

I think the significant attrition rate needs to be mentioned in the abstract, especially as the emphasis in this paper is 12 month follow-up.

It’s difficult to ascertain whether all eligible patients in Australia were invited to take part or not, and whether all prescribing services were recruitment sites. This could be clarified in the methods section.

The results section is rather long and difficult to follow in places, perhaps some of the information could be moved to the supplementary material and summarised in the main text.

“In all cases, average daily doses that were THC-dominant had greater odds of larger improvements in these outcomes compared with THC:CBD-balanced (Table 6).” – is this correct? According to the table, CBD-dominant was also better than balanced for anxiety.

Reviewer #2: Comment 1: The introduction mentions that the study aims to report the 12-month results of QUEST, it only briefly addresses the specific hypothesis near the end. A more focused and explicit research question and objective section would benefit readers, providing a clearer sense of the study's purpose and anticipated contributions to the field.

Comment 2: The target sample size of 2142 is stated, the rationale behind this figure is not clear. Providing more details on how this number was calculated—such as the expected effect size, desired statistical power, and variability in the outcome measures—would clarify why this sample is adequate to achieve robust findings.

Comment 3: The criteria for patient inclusion (e.g., age, prescription of specific products) are clear, but the rationale for excluding those who accessed MC within the previous four weeks or received end-of-life palliative care could use more justification. Clarifying how these exclusions impact the study’s generalizability would enhance the methodological rigor.

Comment 4: Given the real-world variation in THC ratio and dose regimen, further exploration of how different dosing regimens could impact outcomes would be valuable. This would provide more practical guidance for clinicians.

Comment 5: The discussion could benefit from a more detailed future research section, especially concerning the optimal CBD ratios for specific conditions, dose timing, and the need for larger, multicenter RCTs to confirm these findings.

**Do you want your identity to be public for this peer review?** For information about this choice, including consent withdrawal, please see our Privacy Policy

Reviewer #1: No

Reviewer #2: No

---

## [Author Response · Author response to Decision Letter 1]

23 Dec 2024

23 Dec 2024

Francesca Baratta

Academic Editor

PLOS ONE

Dear Dr Baratta,

We thank you and the reviewers for your positive and constructive comments on our manuscript entitled ‘Improvements in health-related quality of life are maintained long-term in patients prescribed medicinal cannabis in Australia: The QUEST Initiative 12-month follow-up observational study.’

It was valuable to be able to revisit the manuscript and we feel that by addressing the reviewer comments, we have strengthened our manuscript. Below we describe the ways in which reviewers’ comments have been incorporated into the manuscript. The revised manuscript has been uploaded along with a marked-up copy.

We look forward to your favourable consideration.

Kind regards,

Margaret-Ann Tait

1. Please ensure that your manuscript meets PLOS ONE's style requirements, including those for file naming…

We have checked that the title page, headings, figures, and file names follow the PLOS ONE manuscript style templates.

2. When completing the data availability statement of the submission form, you indicated that you will make your data available on acceptance…

The relevant deidentified dataset and data dictionary to replicate our study findings have been uploaded to the University of Sydney hosted Sydney eScholarship repository, and is publicly available via this link:

https://url.au.m.mimecastprotect.com/s/7U2CCzvkyVC84v8AOs4gfPO?domain=hdl.handle.net

3. Please review your reference list to ensure that it is complete and correct. If you have cited papers that have been retracted, please include the rationale for doing so in the manuscript text, or remove these references and replace them with relevant current references…

We have reviewed the reference list and checked each title against The Retraction Watch Database, and confirm the list is complete and correct.

Reviewers’ comments:

Reviewer #1:

Thanks for inviting me to review this paper exploring patient reported outcomes following medical cannabis treatment. I have a few comments:

I think the significant attrition rate needs to be mentioned in the abstract, especially as the emphasis in this paper is 12 month follow-up.

We thank the reviewer for their comments and the opportunity improve the presentation of results. We have updated the abstract results to include the questionnaire completion rate at 12-months:

“Of 2744 consenting participants who completed baseline assessments, 2353 also completed at least one follow-up questionnaire and were included in analyses, with completion rates declining to 778/2353 (38%) at 12-months…”

It’s difficult to ascertain whether all eligible patients in Australia were invited to take part or not, and whether all prescribing services were recruitment sites. This could be clarified in the methods section.

Thank you for drawing our attention to the missing information in our methods section. The following sentence has been added to the Methods section under ‘Study Population and design’:

“… All clinicians prescribing Little Green Pharma (LGP) MC oil products across Australia were informed of the study and invited to contact the researchers to receive the study training and information required to screen their patients for eligibility…”

Please note that the preceding paragraph states “Full details of study design, eligibility, recruitment procedures, and data collection are reported in the published study protocol.”

The results section is rather long and difficult to follow in places, perhaps some of the information could be moved to the supplementary material and summarised in the main text.

Thank you for the suggestion. To reduce the main text, we have moved Table 2 to the supplementary material (S3_Table). We felt that all other information should be presented in the main text, however, we are open to moving more results if the editor has identified specific content as being more suitable as supplementary material.

“In all cases, average daily doses that were THC-dominant had greater odds of larger improvements in these outcomes compared with THC:CBD-balanced (Table 6).” – is this correct? According to the table, CBD-dominant was also better than balanced for anxiety.

Thank you for finding this possibly misleading omission in our results description. We have updated the sentence to:

“…, although CBD-dominant was also better than THC:CBD-balanced for anxiety.”

Reviewer #2:

Comment 1: The introduction mentions that the study aims to report the 12-month results of QUEST, it only briefly addresses the specific hypothesis near the end. A more focused and explicit research question and objective section would benefit readers, providing a clearer sense of the study's purpose and anticipated contributions to the field.

We have updated the third paragraph of the introduction to include the objectives of the QUEST study and to justify the current study assessment at 12-months:

‘It is typical when evaluating clinical care outcomes in chronic conditions to assess 12-month follow-up.’ This study aimed to assess 12-month follow-up data to determine if our previously reported improvements at 3-months were maintained long-term and to explore differences across health conditions and MC compositions.

Comment 2: The target sample size of 2142 is stated, the rationale behind this figure is not clear. Providing more details on how this number was calculated—such as the expected effect size, desired statistical power, and variability in the outcome measures—would clarify why this sample is adequate to achieve robust findings.

Details on sample size calculation were reported in the protocol as cited: “Our target sample size of 2142 was determined a priori and reported in the study protocol.[22]” To address the reviewer’s comments, we have updated this sentence with additional information:

‘Our target sample size of 2142 was determined a priori with power to detect the smallest QLQ-C30 effect size threshold [28] using a two-sided significance level of 1%, as reported in the study protocol.[23]’

Comment 3: The criteria for patient inclusion (e.g., age, prescription of specific products) are clear, but the rationale for excluding those who accessed MC within the previous four weeks or received end-of-life palliative care could use more justification. Clarifying how these exclusions impact the study’s generalizability would enhance the methodological rigor.

To address the reviewer’s suggestions, the following information has been added to the Methods section under ‘Study Population and design’:

‘To achieve a pre-therapy baseline, patients were excluded if they had accessed prescribed MC within the previous 4-weeks; selected because it ensured the minimum wash-out period of 13-30 days had passed,[25, 26] and was greater than the maximum recall period of PROMs used in the study. Palliative care patients were identified by clinicians following the ICD-11 definition of having a life expectancy of only a few months.[27] Accordingly, PROMs were only administered to palliative care patients for the first 3-months of the QUEST study, excluding them from the 12-month analysis. Our 3-month findings for participants receiving end of life palliative care are reported elsewhere.[20]’

Comment 4: Given the real-world variation in THC ratio and dose regimen, further exploration of how different dosing regimens could impact outcomes would be valuable. This would provide more practical guidance for clinicians.

We believe clinicians can make inferences from the information provided in table 6 which shows the various THC/CBD dosage ratios and their associations with patient-reported outcomes. As an observational study, we cannot comment on the efficacy of dosage regimens, or medicinal cannabis in general, and our findings cannot be used as clinical guidance. We have addressed this further in the discussion:

‘Our findings suggest there are clinically important differences in outcomes for some conditions depending on the ratio of THC and CBD, however as an observational study, we cannot infer efficacy of different dosage regimens conclusively…’

Comment 5: The discussion could benefit from a more detailed future research section, especially concerning the optimal CBD ratios for specific conditions, dose timing, and the need for larger, multicenter RCTs to confirm these findings.

We thank the reviewer for this suggestion and agree. We have updated the following sentence on future research to the discussion:

‘… As MC becomes increasingly used and accepted globally, further research with multicentre RCTs is needed in this area to provide clinicians with evidence-based guidance on condition-specific MC prescribing and the efficacy of various dosing regimens for their patients.’

---

## [Decision Letter · Decision Letter 1]

25 Feb 2025

Improvements in health-related quality of life are maintained long-term in patients prescribed medicinal cannabis in Australia: The QUEST Initiative 12-month follow-up observational study.

PONE-D-24-17275R1

Dear Dr. Tait,

We’re pleased to inform you that your manuscript has been judged scientifically suitable for publication and will be formally accepted for publication once it meets all outstanding technical requirements.

Kind regards,

Francesca Baratta, PharmD, PhD

Academic Editor

PLOS ONE

Reviewer's Responses to Questions

**Comments to the Author**

Reviewer #1: All comments have been addressed

2. Is the manuscript technically sound, and do the data support the conclusions?

Reviewer #1: Yes

3. Has the statistical analysis been performed appropriately and rigorously?

Reviewer #1: Yes

4. Have the authors made all data underlying the findings in their manuscript fully available?

Reviewer #1: No

5. Is the manuscript presented in an intelligible fashion and written in standard English?

Reviewer #1: Yes

Reviewer #1: Thanks for addressing my comments, I have no further queries

**Do you want your identity to be public for this peer review?** For information about this choice, including consent withdrawal, please see our Privacy Policy

Reviewer #1: No

---

## [Editor Report · Acceptance letter]

PONE-D-24-17275R1

PLOS ONE

Dear Dr. Tait,

I'm pleased to inform you that your manuscript has been deemed suitable for publication in PLOS ONE. Congratulations! Your manuscript is now being handed over to our production team.

Kind regards,

on behalf of

Dr. Francesca Baratta

Academic Editor

PLOS ONE